# DHENN: A Deeper Hybrid End-to-end Neural Network for Highly Accurate Drug-Drug Interaction Events Prediction

## Abstract

Accurate prediction of drug-drug interactions (DDIs) is crucial for therapeutic safety yet poses a substantial challenge due to complex pharmacodynamics. Traditional DDI prediction methods often falter for three reasons. First, they simplify dependency structures among entities (e.g., drugs, targets, enzymes, and transporters) in bipartite networks, falling short in modeling drug-centered high-order information. Second, the over-smoothing effects constrain the depth of the adopted neural networks, thereby limiting their learning capacity. Third, they either partially consider drug-centered relationships or do not unify multiple drug-centered relationships into an end-to-end learning model. In response, this paper proposes *Deeper Hybrid End-to-end Neural Network* (DHENN), which integrates a Multimodal Knowledge Graph (MKG) with a Prediction-Enhanced Cascading Network (PECN) in an end-to-end learning manner. Specifically, MKG captures higher-order information across drug-centered entities, offering a holistic view of DDIs. PECN mitigates over-smoothing associated with feature extraction by incorporating shallow embeddings into deeper layers, preserving node-level diversity. The end-to-end learning manner guarantees that the representation learning and predictive modeling of MKG and PECN are formulated into a unified learning objective. Extensive experiments substantiate that DHENN outperforms thirteen competitors on two real-world DDI datasets.

## Introduction

Drug-Drug Interactions (DDIs) arise from complex pharmacodynamics, where one drug may alter in-vivo behaviors (e.g., serum concentration) of its partners when taken together (Zitnik et al., 2018). Unknown DDIs among multiple administrated drugs in clinical settings can result in accidental adverse reactions, some of which are literally deadly (Leape et al., 1995). Accurate prediction of unknown DDIs events are gaining prominence for clinical safety, given the rising costs of vitro experiments and concerns for animal welfare. Predictive modeling for DDI prediction can be traced back to seminal work by (Prichard & Shipman Jr, 1990) and has since spurred an flurry of studies, e.g., (Huang et al., 2020; Ryu et al., 2018; Cui et al., 2020; Lin et al., 2020; Xiong et al., 2023).

Topological structures provide a natural depiction of DDI events through a DDI network, where each link connecting nodes (drugs) represents their interactions. Predicting DDIs in these networks involves determining the existence (binary) or category (multi-class) of these links (Abbas et al., 2021). Efforts to enrich DDI studies have focused on incorporating multifaceted topological information to enhance network content. For example, nodal contents are enriched by learning vector representations from molecular structures (Feng & Zhang, 2022; Yu et al., 2022), clinic side-effect reports (Iyer et al., 2014), and drug-food constituencies (Ryu et al., 2018). To respect biological dynamics between drug and protein (e.g., target, enzyme, transporter) (Cui et al., 2020), recent research leverages various bipartite graphs to describe the drug-protein interactions and use graph neural networks (GNNs) for feature fusion (Deng et al., 2020; Lin et al., 2022a;b; Tang et al., 2024).

Despite advancements, most DDI prediction studies boil down to inherent the link prediction paradigm, which may lead to inferior modeling precision because of three challenges. First, the current DDI networks are tailored to model first-order, bipartite relationships, such as <drug, transporter> and <drug, enzyme>. Thus, they struggle to delineate high-order information linked

by drugs, such as $<$ drug, transporter, drug, enzyme, drug, molecular structures$>$. Such high-order pathways convey pharmacological details of how drugs are absorbed, distributed, metabolized, and excreted, thereby establishing a more holistic view of DDI events.

Second, the current backbone neural networks for DDI prediction are mostly shallow. This is because of *over-smoothing* (Chen et al., 2020; Liu et al., 2020), where all drug embeddings tend to be indistinguishable after multiple layers of representation. Shallow networks in general cannot afford enough learning capacity to capture complex drug-centered high-order information.

Third, there are rare existing DDI models that comprehensively consider drug-centered multiple relationships. Besides, the rare ones do not unify multiple drug-centered relationships into an end-to-end learning model. Such a learning way decouples the stages of representation and prediction, leading to suboptimal solutions that overlook the potential for ground-truth DDIs to refine embedding generation. into training the DDI prediction model. As a result, they obtain sub-optimal embeddings of by overlooking the fact that ground-truth DDIs can in turn refine the process of embedding generation. Specifically, they mainly learn drug embeddings from multiple drug-centered relationships at first and then feed the well-learned embeddings into a DDI predictor for training. As a result, they obtain sub-optimal embeddings by overlooking the fact that ground-truth DDIs can in turn refine the embedding process.

Motivated by the status quo, this study proposes a novel *Deeper Hybrid End-to-end Neural Network* (DHENN) model. Our DHENN is designed based on three key ideas. First, to capture higher-order information, we construct a Multimodal Knowledge Graph (MKG) that connects various types of entities related to DDI events (e.g., drugs, targets, enzymes, transporters, molecular substructures) in one topology. Second, to enlarge learning capacity, we design a Prediction-Enhanced Cascading Network (PECN) to dynamically combine shallow node embeddings into the subsequent representation layers. Third, DHENN couples its representation learning and predictive modeling stages in an end-to-end way, where the feature extractions and fusions from raw entity modalities to ground-truth DDIs are formulated into a unified learning objective. As such, DHENN can stack *deep* hidden layers to learn higher-order and deeper latent features, as well as guarantee the feature extractions and fusions to be optimal.

**This paper has the following specific contributions:**

- We propose a highly accurate DHENN model for predicting DDI events, which exploit the high-order information between various types of entities related to various DDI events in one topology of MKG.
- This is the first study to design a *deep* PECN to learn deeper latent features from MKG of DDI events in an end-to-end fashion.
- Extensive experiments on real drug datasets are conducted to evaluate our DHENN model. The results demonstrate that DHENN exhibits high accuracy and significantly outperforms nine state-of-the-art and four traditional models in predicting DDI events.

## RELATED WORK

This section notes the recent advances in DDI predictive modeling using graphs. A more comprehensive literature survey including non-graph DDI models is deferred to the **APPENDIX** due to page limits.

**Single graph-based DDI prediction.**   These methods rely on a homogeneous DDI network only, casting DDI prediction as a link prediction task. They can be categorized in three groups. Namely, 1) matrix factorization that aims to complete the DDI adjacency matrix (Shi et al., 2019), 2) random walk that generates node embeddings from sequences to calculate their similarities (Ribeiro et al., 2017), and 3) hard-encoded graph feature extraction that predefines topological patterns such as centrality or connectivity for node embeddings (Tang et al., 2015; Wang et al., 2016).

**Dual graph-based DDI prediction.**   These methods integrate two graphs of DDI and molecular interactions, predicting DDI events and molecular properties at once (Wang et al., 2022; Li et al., 2022). In particular, MRGNN (Xu et al., 2019) employs multiple graph convolution layers to extract node features from diverse neighboring nodes within a structured entity graph. MFFGNN (He et al., 2022) integrates the topological structure within molecular graphs with the interaction relationship

between drugs, along with the local chemical context encoded in SMILES sequences. EPGCN-DS (Sun et al., 2020) adopts a framework based on graph convolutional networks for type-specific DDI identification from molecular structures. In addition, Molormer (Zhang et al., 2022) leverages the 2D structures of drugs as input and uses a lightweight attention mechanism to encode the spatial information of the molecular graph.

**Knowledge graph-based DDI prediction.**  Knowledge graphs (KGs) enable a more holistic view of DDI modeling by integrating multiple types of biological entities and relations, including drugs, targets, enzymes, and transporters. KGNN (Lin et al., 2020) integrates graph convolutional networks with neighborhood sampling, effectively extracting valuable neighborhood relations. AAEs (Dai et al., 2020) uses KG embedding through adversarial autoencoders, along with Wasserstein distances and GumbelSoftmax relaxation, to enhance the learning process. SumGNN (Yu et al., 2021) proposes a graph summarization module designed for subgraphs, allowing the extraction of meaningful pathways that can be easily managed and analyzed. In a similar vein, LaGAT (Hong et al., 2022) leverages a link-aware graph attention method that generates multiple attention pathways for drug entities based on the diverse links between drug pairs. DDKG (Su et al., 2022) furthers these efforts by learning drug embeddings from their attributes within KGs and incorporating neighboring node embeddings and triple facts simultaneously. EmerGNN (Zhang & Yao, 2023) predicts interactions for emerging drugs by leveraging the rich information in biomedical networks. MKG-FENN (Wu et al., 2024) adopts a comprehensive and end-to-end framework to achieve optimal feature extraction and fusion. KnowDDI (Wang & Yang, 2024) enhances drug representations by adaptively leveraging rich neighborhood information from large biomedical knowledge graphs.

**Hybrid graph and feature extraction modeling.**  There are studies combining graphs with nodal feature extraction in various DDI models, which often lead to better prediction performances over individual models (Chen et al., 2021). For example, MDNN (Lyu et al., 2021) combines a drug knowledge graph pathway with a heterogeneous features pathway, and MIRACLE (Wang et al., 2021) uses contrastive learning that treats a DDI network as a multiview graph with each node representing an individual drug molecular instance. Deepika & Geetha (2018) employs a semi-supervised learning framework that incorporates network representation learning and meta-learning techniques. GoGNN (Wang et al., 2020) uses a dual-attention mechanism to extract hierarchical features from structured entity graphs and DDI networks. MUFFIN (Chen et al., 2021) is a multi-scale feature fusion model that combines drug structure and a biomedical KG for improving drug node embedding. MRCGNN (Xiong et al., 2023) integrates the features of DDI events and drug molecular graphs by GNNs.

**Novelty.**  We note three unique differences between previous methods and our proposal. First, the graph-based methods commonly separate the drug-centered binary relations and thus ignore high-order information that may be linked through intermediate entities, e.g., a drug-enzyme-drug pathway. Second, these methods are limited by the over-smoothing effect in shallow neural networks. Third, these methods either partially consider drug-centered relationships or do not unify drug-centered multiple relationships into an end-to-end learning model. In contrast, our DHENN model enjoys three merits: 1) exploiting the high-order information from various drugs, chemical entities, and molecular structures by unifying them into one MKG topology, 2) designing a deeper PECN to mitigate the over-smoothing associated with the nodal feature extraction on MKG, and 3) guaranteeing the feature extractions and fusions to be optimal by an end-to-end learning way.

## PRELIMINARIES

**DDI Matrix.**  The DDI matrix serves as a representation of drug-drug interaction occurrences and is denoted as $\mathcal{Y} \in (0, y_{ij})^{N_d \times N_d}$, where $N_d$ represents the number of drugs included. Each element $\mathcal{Y} \in (0, y_{ij})$ in the matrix indicates the presence or absence of a drug interaction event between drug $d_i$ and drug $d_j$. If $y_{ij} = 0$, it signifies the absence of an interaction event between the two drugs, while any other value indicates the presence of an interaction event. By utilizing the label matrix, researchers can characterize different types of drug-drug interactions. The label set $\mathcal{L} = \{y_1, y_2, \cdots, y_{N_l}\}$ represents the possible labels, with $N_l$ denoting the number of event types. Each element $y_{ij} \in \mathcal{L}$ in the DDI matrix represents a specific label, providing information about the nature of the interaction between drug $d_i$ and drug $d_j$.

**Drug Knowledge Graph.**  The drug knowledge graph, denoted as $\mathcal{G} = (\mathcal{D}, \mathcal{R}, \mathcal{T})$, is a specialized knowledge graph designed for predicting drug-drug interaction events. It consists of three compo-

nents: $\mathcal{D}$, representing a subset of drug entities; $\mathcal{R}$, representing the set of relations between drugs and tail entities; and $\mathcal{T}$, representing a subset of tail entities related to drugs (e.g., targets). The drug knowledge graph is defined as a collection of tuples $(d, r_{dt}, t)$, where each tuple represents a connection between a drug entity $d$, a relation $r_{dt}$, and a tail entity $t$. These connections exist if and only if the drug entity is in the set $\mathcal{D}$, the relation is in the set $\mathcal{R}$, and the tail entity is in the set $\mathcal{T}$. By analyzing the drug knowledge graph, valuable insights can be gained regarding the relationships between drugs and their associated tail entities, providing valuable information for predicting DDIs.

**DDI Events Prediction.** Our primary objective is to predict specific drug interaction events between drug $d_i$ and drug $d_j$ using both the DDI events matrix $\mathcal{Y}$ and the drug knowledge graph $\mathcal{G}$. To accomplish this task, we employ a prediction function denoted as $\hat{y}_{ij} = \Gamma\left(d_i, d_j \mid \Theta, \mathcal{Y}, \mathcal{G}\right)$. This function combines model parameters $\Gamma$ with the information from $\mathcal{Y}$ and $\mathcal{G}$ to provide reliable predictions of the occurrence of interaction events between drug $d_i$ and drug $d_j$. By considering multiple factors and leveraging the available data, our approach aims to enhance the accuracy and effectiveness of DDI events prediction.

## PROPOSED METHOD

**Overview.** The overall structure of our proposed DHENN model is illustrated in Figure 1. DHENN is primarily divided into two parts. In the upper part, a graph neural network (GNN) is to extract higher-order and semantic features from a constructed multimodal knowledge graph (MKG). In the lower part, a prediction-enhanced cascading network (PECN) is designed to integrate the extracted features and predict the types of DDIs. By combining these two parts, DHENN can effectively analyze and predict DDIs based on the learned features and relationships.

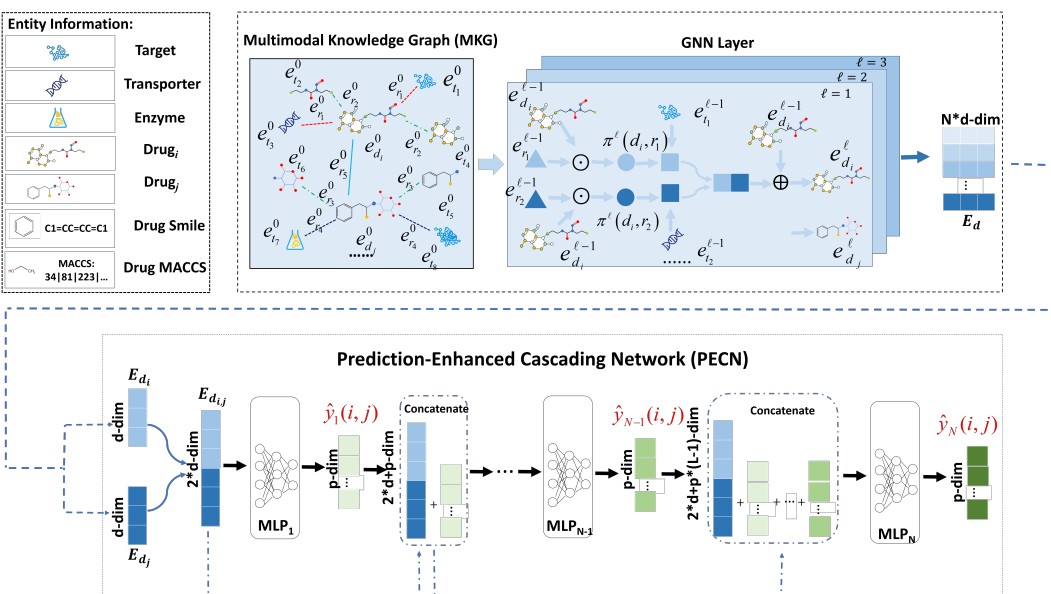

Figure 1: The overall structure of the proposed DHENN model.

## FEATURE EXTRACTION AND FUSION IN MKG

### MKG CONSTRUCTION.

As depicted in Figure 1, our MKG (i.e., drug knowledge graph) is a complex network that provides a clear description of the intricate semantic relationships between drugs and molecular structures, chemical entities, substructures, and other drugs. The drug knowledge graph can be described in the form of tuples as <drugs, relationships, entities>. To better understand how drugs are related to entities, we provide a detailed explanation of the relationships between drugs and various types of entities.

**Drugs-chemical Entities.** We gather drug-related information, including transporters and targets, to serve as the entities. We assign the corresponding relationship based on the general function of the entity. For example, let's take the drug Lovastatin. If there is a transporter named Serum albumin, and its general function is Toxic substance binding, we would create the following triplet: <Lovastatin, Toxic substance binding, Serum albumin>.

**Drug-substructures.** The SMILES attribute of drugs is treated as entities, and the relationship between drugs and entities is represented by "including". For example, if the drug Lovastatin has a SMILES attribute "986", the corresponding triplet would be <Lovastatin, including, 986>.

**Drugs-drugs.** The DDI events matrix is renowned for its extensive scale and rich information. Within this dataset, we can gather information about the other drugs that each drug can interact with. We treat these drugs as entities, and the specific interaction events as the corresponding relationships. For example, Abemaciclib interacting with Bosutinib leads to an increase in serum concentrations. Therefore, the corresponding triplet would be <Abemaciclib, increase in serum concentrations, Bosutinib>.

**Molecular structures.** This portion of the data is based on the Molecular ACCess System (MACCS) bonds, along with 13 MACCS bonds and 7 other molecular features. These MACCS bonds and molecular features are considered as entities of the drug, where the values indicating their occurrence frequencies are denoted as relationships. For example, Glucosamine has three occurrences of the molecular substructure "NumSaturatedRings," represented as <Glucosamine, 3, NumSaturatedRings>.

It is worth noting that the Unified Medical Language System (UMLS) and the DrugBank ID are utilized as a unified identifier system to construct our MKG.

### MKG LEARNING VIA GRAPH NEURAL NETWORK.

The objective of employing the GNN layer is to capture the topological structure and semantic relationships inherent in drugs. In this paper, the drug knowledge graph is converted into a matrix representation. The initial representation matrix of the drug knowledge graph, denoted as $E_{\mathcal{G}}$ can be expressed in the following format:

$$
E_{\mathcal{G}} = [\underbrace{e_{d_1}^{(0)}, \cdots, e_{N_d}^{(0)}}_{\text{drug-embedding}}, \underbrace{e_{r_1}^{(0)}, \cdots, e_{N_r}^{(0)}}_{\text{relation-embedding}}, \underbrace{e_{t_1}^{(0)}, \cdots, e_{N_k}^{(0)}}_{\text{tail-embedding}}],
\tag{1}
$$

where $E_{\mathcal{G}}$ represents the initial representation matrix of the knowledge graph. The variables $N_d$, $N_r$, and $N_k$ indicate the number of drugs, relationships, and tail entities, respectively. The embeddings $e_d^{(0)} \in R^d$, $e_r^{(0)} \in R^d$ and $e_t^{(0)} \in R^d$ represent the initial embeddings for drugs, relationships, and tail entities, respectively. These embeddings are vectors in the $d$-dimensional space, where $d$ is the embedding dimension of the drug knowledge graph.

To capture the neighborhood information of each drug $d_i$, a fixed-size sample of neighbors is uniformly selected instead of considering all tail entities. These sampled neighbors are denoted as $N_s(d_i)$, representing the fixed-size neighborhoods associated with drug $d_i$. The sampled neighbors can be described using the following formula:

$$
N_s(d_i) = \begin{cases} \text{Rand}\left(e_{t_n}^{(0)}, \text{replace} = \text{False}\right), & \text{NS} >= \text{TS} \\ \text{Rand}\left(e_{t_n}^{(0)}, \text{replace} = \text{True}\right), & \text{NS} < \text{TS} \end{cases}
\tag{2}
$$

When the overall neighbors of a drug is greater than or equal to a fixed sampling neighbors, the $Rand$ function will non-repetitively select a fixed neighbors. $TS$ represents the size of the overall neighborhood, while $NS$ represents the size of the adopted neighborhood. However, when the overall neighbors is smaller than the fixed sampling neighbors, we will allow it to repetitively select a fixed neighborhood.

For the drug $d_i$ in the drug knowledge graph $\mathcal{G}$, we represent it using triples $(d_i, r_{in}, t_n)$, where $t_n$ represents the neighborhood of drug $d_i$, and $r_{in}$ represents the semantic relationship within that neighborhood. In order to incorporate the semantic information of relationships into the learning of drug representations, we calculate the semantic feature score between drug $d_i$ and its corresponding neighborhood tail entity $t_n$ using the following formula:

$$\pi_{(d_i, r_{in})}^{(l)} = \text{sum} \left[ \left( e_{d_i}^{(l-1)} \odot e_{r_{in}}^{(l-1)} \right) W_l^{(p)} + b_l^{(p)} \right] \tag{3}$$

In the formula, $e_{r_{in}}^{(l-1)}$ represents the embedding of the relationship between drug $d_i$ and tail entity $t_n$ in the $(l-1)^{th}$ layer of the GNN. $e_{d_i}^{(l-1)}$ represents the embedding of drug $d_i$ in the $(l-1)^{th}$ layer of the GNN. $W_l^{(p)}$ denotes the trainable weight matrix, $b_l^{(p)}$ represents the bias vector, and $p$ signifies the number of fully connected layers. The symbol $\odot$ denotes element-wise multiplication.

Next, we perform aggregation on the embeddings of the neighborhood $N_s(d_i)$ by combining them with the corresponding semantic feature scores. The aggregation function is defined as:

$$e_{N_s(d_i)}^{(l)} = \sum_{l_n \in N_s(d_i)} \pi_{(d_i, r_{in})}^{(l)} e_{t_n}^{(l-1)} \tag{4}$$

In the formula, $e_{t_n}^{(l-1)}$ denotes the neighborhood embedding associated with drug $d_i$ within the $(l-1)^{th}$ layer of the GNN. On the other hand, $\pi_{(d_i, r_{in})}^{(l)}$ signifies the semantic feature score corresponding to drug $d_i$ and its relationship within the $(l)^{th}$ layer.

The next step involves the aggregation process. To amalgamate the drug $d_i$ embedding alongside its associated neighborhood representation into a vector, we employ the fusion equation:

$$E_{d_i} = e_{d_i}^{(l)} = \sigma \left( \left( e_{d_i}^{(l-1)} \oplus e_{N_s(d_i)}^{(l)} \right) W_2 + b_2 \right) \tag{5}$$

Finally, in order to maximize the information from the drug, we use the above calculation method to obtain the representations of different categories of entities for drugs. Then we concatenate the drug representations of different categories of entities together to obtain the final drug representation.This allows us to capture a comprehensive view of the drug by incorporating various relevant features.The fusion of the representations of different aspects of the drug can be described using the following formula:

$$\hat{E}_{d_i} = E_{d_i}^1 \oplus E_{d_i}^2 \oplus \cdots \oplus E_{d_i}^n \tag{6}$$

In the formula, we use $\hat{E}_{d_i}$ to represent the final representation of drug $d_i$, and $E_{d_i}^1$, $E_{d_i}^2$ and $E_{d_i}^n$ represent the first,second and $nth$ category of drug representations, respectively.

Likewise, we can utilize the same approach to compute the representation of drug $d_j$ by leveraging its respective knowledge graphs. By employing the formula and generating the drug representation, we can effectively capture the pertinent information and features associated with drug $d_j$ within the MKG.

PREDICTION-ENHANCED CASCADING NETWORK

To enhance learning of complex DDI patterns, PECN in DHENN dynamically merges shallow embeddings into deeper layers. PECN uses a cascaded structure, where each layer takes predicted output from the previous layer as input, with each layer being an MLP architecture.

In the first layer, we concatenate the drug representations of $d_i$ and $d_j$ from MKG as the input. Both $d_i$ and $d_j$ have a dimension of $d$, so when concatenated, they form a $2*d$-dimensional vector. The output is a $c$-dimensional vector representing the predicted category of the DDI events. The prediction formula for the first layer can be expressed as:

$$\hat{y}_1(i,j) = \sigma \left( \left( \hat{E}_{d_i} \oplus \hat{E}_{d_j} \right) W_3^{(q)} + b_3^{(q)} \right) \tag{7}$$

In the subsequent layers of PECN (N layers), the inputs are formed by concatenating the drug representations of $d_i$ and $d_j$ along with the output from the previous layers. The dimensionality of the input in this layer is $2*d + c*(N-1)$, where $N$ represents the layer number, which is greater than 1. The formula for the prediction in these layers can be expressed as:

$$\begin{aligned} \hat{y}_N(i,j) = & \sigma \left( \left( \hat{E}_{d_i} \oplus \hat{E}_{d_j} \oplus \hat{y}_1(i,j) \oplus \cdots \right. \right. \\ & \left. \left. \oplus \hat{y}_{N-1}(i,j) \right) W_{N+2}^{(q)} + b_{N+2}^{(q)} \right) \end{aligned} \tag{8}$$

To ensure the objective of the loss function aligns effectively with the learning parameters, we employ a hybrid loss function that directly influences the parameter learning of the corresponding MLP layer. This approach allows DHENN to optimize the learning process by incorporating the relevant information from the loss function into the MLP layer's parameter updates. The loss function of the model can be represented using the following formula:

$$Loss = \alpha_1 CrossEntropyLoss\left(\hat{y}_1(i,j), y\right) + \cdots \\ + \alpha_n CrossEntropyLoss\left(\hat{y}_n(i,j), y\right) \tag{9}$$

In this cascaded loss function, each sub-loss function corresponding to each MLP is cross-entropy, where $\alpha_1$ and $\alpha_n$ represent the weights of the first and $N$-th cross-entropy, respectively. To optimize, we integrate a batch normalization layer to accelerate convergence. Moreover, a dropout layer and $\ell_2$ regularization are used to alleviate overfitting.

## ILLUSTRATIVE EXAMPLE OF OUR DHENN

This subsection uses an illustrative example to explain the model methodology, as shown in Figure 2. Supposing there is a dataset with 572 drugs, 4 different types of entities, and 65 types of DDI events. The first is to construct an MKG that can be represented by a tuple: $<$drugs, chemical entities, substructures, drugs, molecular structures$>$. A fixed-size neighborhood of entities is selected for each category of tail entities, which is illustrated in Figure 2(a). Then, high-order topological information and semantic relationships between drugs and tail entities are

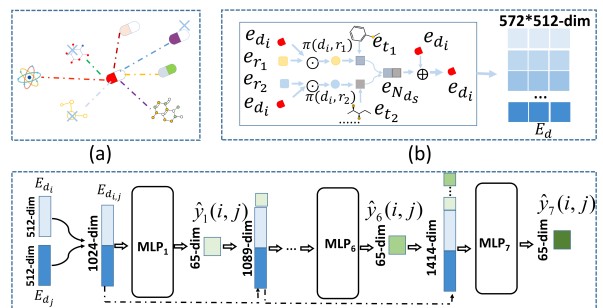

Figure 2: Illustration of our DHENN computational flow.

extracted through GNN layers, as shown in Figure 2(b). If the representation dimension of the drug $d_i$ is 128. Then, by concatenating the representations of the four different types of tail entities, we obtain a final representation of the drug $d_i$ with a length of 512 dimensions. Therefore, a $572 * 512$-dimensional matrix is obtained.

Next, such a matrix is input into the PECN for predicting DDI events as shown in Figure 2(c). By concatenating the drugs $d_i$ and $d_j$ to obtain a 1024-dimensional vector as the input of the first MLP classifier. This results in a 65-dimensional output vector, which corresponds to the number of predicted types of DDI events. In the subsequent MLP classifiers, we take the input vector and output vector of the previous MLP classifier as the input vector of the current classifier. Therefore, in the $N$-th layer MLP classifier, the input vector dimension is $1024 + 65 * (n-1)$ dimensions, and the output is 65 dimensions. If there is a 7-layer MLP classifier, resulting in a 1414-dimensional vector as the input to the final layer of the classifier to obtain the final output of 65-dimensional prediction vectors.

### ALGORITHM DESIGN
Due to the page limitation, the algorithm pseudocode and the complexity analysis of DHENN are moved into the **Appendix**.

### EXPERIMENTS

In the subsequent experiments, three research questions (RQs) are investigated as follows:

- RQ.1. Can the proposed DHENN model outperform state-of-the-art models in predicting DDI events between known and/or new drugs?

- RQ.2. How do the three key ideas (i.e., MKG, FENN, and end-to-end learning manner) of the proposed DHENN model impact its performance (i.e., ablation study)?

- RQ.3. How do the hyper-parameters of the proposed DHENN model impact its performance?

General Settings

**Datasets.** The first dataset (Dataset1) was collected by DDIMDL[1] from DrugBank, and it consists of 572 drugs, 74,528 triple relationships, and is associated with 65 DDI events. Each drug has four entity types: drugs-chemical entities, drug-substructures, drug-drugs, and molecular structures. The second dataset (Dataset2) also originates from DrugBank[2]. It comprises 846 drugs, 92,105 triple relationships, and is associated with 73 DDI events. Each drug has three entity types: drugs-chemical entities, drug-substructures, and drugs-drugs.

**Evaluation Metrics.** To evaluate the proposed DHENN model, a set of multi-class classification evaluation metrics is adopted. These metrics include accuracy (ACC), area under the precision-recall curve (AUPR), area under the ROC curve (AUC), F1 score, precision (Pre), and recall.

**Baselines.** We compare DHENN with nine state-of-the-art related models: MKG-FENN (Wu et al., 2024), EmerGNN (Zhang & Yao, 2023), KnowDDI (Wang & Yang, 2024), MDDI-SCL (Lin et al., 2022a), MDF-SA-DDI (Lin et al., 2022b), DDIMDL (Deng et al., 2020), MDNN (Lyu et al., 2021), Lee et al.'s methods (Lee et al., 2019), and DeepDDI (Ryu et al., 2018). Furthermore, we also consider several traditional methods, including DNN, RF, KNN, and LR (Deng et al., 2020). Please refer to the **Appendix** to see more details

**Hyper-Parameter.** Most of the hyper-parameters were set the same for the two datasets. We simultaneously set 100 epochs of iteration, a learning rate of 0.01, a neighborhood size of 10, and an embedding size of 128. Additionally, the parameters were set to $l = 1$, $p = 2$, and $q = 3$, where $l$ is the number of hidden layers in the GNN. Our empirical study suggested using 1 layer, which aligns with the recent concern of over-smoothing in GNN (Lyu et al., 2021). For different datasets, we have set different batch sizes (Dataset 1=1024 and Dataset 2=2048) and regularization controlling weight (Dataset 1=1e-08 and Dataset 2=1e-10).

Performance Comparison with Baselines (RQ.1)

Three tasks of predicting DDI events are tested: between known drugs (Task 1), between known drugs and new drugs (Task 2), and predicting DDI events among new drugs (Task 3).

| Dataset | Metric | MKG-FENN | Know-DDI | Emer-GNN | MDDI-SCL | MDF-SA-DDI | DDIMDL | MDNN | Lee et al.' methods | DeepDDI | DNN | RF | KNN | LR | DHENN |
|---|---|---|---|---|---|---|---|---|---|---|---|---|---|---|---|
| Dataset 1 | ACC | 0.9409 | 0.9022 | 0.9343 | 0.9378 | 0.9301 | 0.8852 | 0.9175 | 0.9094 | 0.8371 | 0.8797 | 0.7775 | 0.7214 | 0.7920 | 0.9458 |
| | AUPR | 0.9786 | 0.9436 | 0.9771 | 0.9782 | 0.9737 | 0.9208 | 0.9668 | 0.9562 | 0.8899 | 0.9134 | 0.8349 | 0.7716 | 0.8400 | 0.9750 |
| | AUC | 0.9989 | 0.9852 | 0.9989 | 0.9983 | 0.9989 | 0.9976 | 0.9984 | 0.9961 | 0.9961 | 0.9963 | 0.9956 | 0.9813 | 0.9960 | 0.9988 |
| | F1 | 0.8958 | 0.8653 | 0.8069 | 0.8755 | 0.8878 | 0.7585 | 0.8301 | 0.8391 | 0.6848 | 0.7223 | 0.5936 | 0.4831 | 0.5948 | 0.9032 |
| | Pre | 0.9132 | 0.8429 | 0.8400 | 0.8804 | 0.9085 | 0.8471 | 0.8622 | 0.8509 | 0.7275 | 0.8047 | 0.7893 | 0.7174 | 0.7437 | 0.9317 |
| | Rec | 0.8876 | 0.8322 | 0.7926 | 0.8767 | 0.8760 | 0.7182 | 0.8202 | 0.8339 | 0.6611 | 0.7027 | 0.5161 | 0.4081 | 0.5236 | 0.8933 |
| Dataset 2 | ACC | 0.9516 | 0.9034 | 0.9401 | 0.9514 | 0.9423 | 0.9434 | 0.9462 | 0.9368 | 0.8906 | 0.9342 | 0.8396 | 0.8230 | 0.8537 | 0.9560 |
| | AUPR | 0.9867 | 0.9124 | 0.9824 | 0.9864 | 0.9738 | 0.9749 | 0.9842 | 0.9651 | 0.9484 | 0.9802 | 0.9077 | 0.8848 | 0.9129 | 0.9849 |
| | AUC | 0.9994 | 0.9522 | 0.9995 | 0.9991 | 0.9984 | 0.9992 | 0.9992 | 0.9992 | 0.9973 | 0.9991 | 0.9980 | 0.9920 | 0.9981 | 0.9994 |
| | F1 | 0.9181 | 0.9262 | 0.8633 | 0.9147 | 0.8619 | 0.8863 | 0.9123 | 0.8951 | 0.8146 | 0.8441 | 0.6339 | 0.7088 | 0.6499 | 0.9263 |
| | Pre | 0.9307 | 0.8892 | 0.8902 | 0.9254 | 0.8975 | 0.9464 | 0.9443 | 0.9030 | 0.8554 | 0.9308 | 0.7962 | 0.8419 | 0.7787 | 0.9506 |
| | Rec | 0.9100 | 0.8498 | 0.8520 | 0.9096 | 0.8456 | 0.8502 | 0.8903 | 0.8913 | 0.7945 | 0.8620 | 0.5631 | 0.6491 | 0.5954 | 0.9235 |
| Win/Tie/Loss | | 8/1/3 | 12/0/0 | 9/0/3 | 10/0/2 | 11/0/1 | 12/0/0 | 12/0/0 | 12/0/0 | 12/0/0 | 12/0/0 | 12/0/0 | 12/0/0 | 12/0/0 | **146/1/9** |
| Statistic | *p*-value | 0.0068 | 0.0002 | 0.0032 | 0.0046 | 0.0005 | 0.0002 | 0.0002 | 0.0002 | 0.0002 | 0.0002 | 0.0002 | 0.0002 | 0.0002 | - |
| | F-rank | 2.21 | 8.75 | 5.92 | 3.96 | 6.00 | 7.00 | 5.17 | 6.63 | 11.04 | 8.38 | 12.75 | 13.42 | 12.00 | **1.79** |

Table 1: The comparison between DHENN and its competitors in task 1, including the Win/Tie/Loss counts, Wilcoxon signed-ranks test, and Friedman test.

**Comparison Based on Known Drugs**

Task 1 plays a crucial role in DDI events prediction. Task 1 adopts five-fold cross-validation to divide the datasets into five subsets, with four subsets used for training and one subset for testing, repeatedly. Table 1 presents the comparison results. To gain a deeper understanding of these results, we conducted comprehensive statistical analyses, including win/tie/loss analysis, the Wilcoxon signed-ranks test, and the Friedman test (Demsar, 2006). These analyses provide valuable insights into the performance of DHENN compared to the baselines.

---

[1] https://github.com/YifanDengWHU/DDIMDL
[2] https://go.drugbank.com/

| Dataset | Metric | MKG-FENN | Know-DDI | Emer-GNN | MDDI-SCL | MDF-SA-DDI | DDIMDL | MDNN | Lee et al.' methods | DeepDDI | DNN | RF | KNN | LR | DHENN |
|---|---|---|---|---|---|---|---|---|---|---|---|---|---|---|---|
| Dataset 1 | ACC | 0.6805 | 0.6352 | 0.6673 | 0.6767 | 0.6633 | 0.6415 | 0.6495 | 0.6405 | 0.5774 | 0.6239 | 0.5575 | 0.5084 | 0.4670 | 0.6910 |
| | AUPR | 0.7049 | 0.6558 | 0.6778 | 0.6947 | 0.6776 | 0.6558 | 0.6661 | 0.6244 | 0.5594 | 0.6361 | 0.5644 | 0.4955 | 0.4499 | 0.7101 |
| | AUC | 0.9673 | 0.9437 | 0.9447 | 0.9634 | 0.9497 | 0.9799 | 0.9516 | 0.9247 | 0.9575 | 0.9796 | 0.9669 | 0.8504 | 0.9639 | 0.9725 |
| | F1 | 0.5394 | 0.5558 | 0.5269 | 0.5304 | 0.5584 | 0.4460 | 0.4471 | 0.5039 | 0.3416 | 0.2997 | 0.1679 | 0.2058 | 0.1739 | 0.5351 |
| | Pre | 0.6063 | 0.5533 | 0.6255 | 0.6254 | 0.6547 | 0.5607 | 0.5582 | 0.5388 | 0.3630 | 0.4237 | 0.4722 | 0.3146 | 0.2484 | 0.6413 |
| | Rec | 0.5106 | 0.4351 | 0.4880 | 0.4814 | 0.5078 | 0.4319 | 0.4611 | 0.4891 | 0.3890 | 0.2840 | 0.1313 | 0.1673 | 0.1470 | 0.5173 |
| Dataset 2 | ACC | 0.7100 | 0.6732 | 0.6367 | 0.6866 | 0.6664 | 0.7267 | 0.7255 | 0.6083 | 0.6336 | 0.6838 | 0.5356 | 0.5892 | 0.5610 | 0.7361 |
| | AUPR | 0.7498 | 0.6890 | 0.6983 | 0.7059 | 0.6776 | 0.7526 | 0.7428 | 0.6121 | 0.6283 | 0.7077 | 0.5987 | 0.6115 | 0.5664 | 0.7674 |
| | AUC | 0.9783 | 0.9627 | 0.9729 | 0.9663 | 0.9637 | 0.9871 | 0.9661 | 0.9701 | 0.9350 | 0.9690 | 0.9771 | 0.8879 | 0.9752 | 0.9809 |
| | F1 | 0.5873 | 0.5811 | 0.4442 | 0.5821 | 0.5861 | 0.5794 | 0.6186 | 0.4478 | 0.4905 | 0.5637 | 0.2736 | 0.3128 | 0.3200 | 0.5881 |
| | Pre | 0.6809 | 0.6807 | 0.5177 | 0.6672 | 0.7041 | 0.7578 | 0.7115 | 0.4394 | 0.5455 | 0.6920 | 0.5865 | 0.4358 | 0.4551 | 0.7308 |
| | Rec | 0.5394 | 0.5211 | 0.4673 | 0.5408 | 0.5248 | 0.5044 | 0.5721 | 0.4715 | 0.4666 | 0.5011 | 0.2043 | 0.2658 | 0.2707 | 0.5456 |
| Win/Tie/Loss | | 11/0/1 | 11/0/1 | 12/0/0 | 12/0/0 | 10/0/2 | 9/0/3 | 10/0/2 | 12/0/0 | 12/0/0 | 12/0/0 | 12/0/0 | 12/0/0 | 12/0/0 | **147/0/9** |
| Statistic $p$-value | | 0.0012 | 0.0012 | 0.0002 | 0.0002 | 0.0105 | 0.0081 | 0.0212 | 0.0002 | 0.0002 | 0.0005 | 0.0002 | 0.0002 | 0.0002 | - |
| F-rank | | 3.25 | 3.25 | 4.92 | 4.92 | 5 | 4.17 | 4.83 | 7.83 | 8.83 | 6.75 | 9.58 | 10.92 | 10.17 | **1.75** |

Table 2: The comparison results between DHENN and its competitors in task 2, where partial drugs are unknown in DDIs graphs during training.

| Dataset | Metric | MKG-FENN | Know-DDI | Emer-GNN | MDDI-SCL | MDF-SA-DDI | DDIMDL | MDNN | Lee et al.' methods | DeepDDI | DNN | RF | KNN | LR | DHENN |
|---|---|---|---|---|---|---|---|---|---|---|---|---|---|---|---|
| Dataset 1 | ACC | 0.4552 | 0.4498 | 0.4783 | 0.4589 | 0.4338 | 0.4075 | 0.4575 | 0.4097 | 0.3602 | 0.4087 | 0.3329 | 0.3057 | 0.3126 | 0.4824 |
| | AUPR | 0.4162 | 0.3986 | 0.4411 | 0.3938 | 0.3873 | 0.3635 | 0.4215 | 0.3184 | 0.2781 | 0.3776 | 0.2640 | 0.2223 | 0.2532 | 0.4513 |
| | AUC | 0.9149 | 0.9078 | 0.9201 | 0.9053 | 0.8630 | 0.9512 | 0.8753 | 0.8302 | 0.9059 | 0.9550 | 0.9143 | 0.7332 | 0.9342 | 0.9305 |
| | F1 | 0.2186 | 0.2270 | 0.2127 | 0.1919 | 0.2329 | 0.1590 | 0.1697 | 0.2022 | 0.1373 | 0.1152 | 0.0173 | 0.0468 | 0.0539 | 0.2252 |
| | Pre | 0.2754 | 0.2765 | 0.2873 | 0.2585 | 0.2715 | 0.2408 | 0.2184 | 0.2216 | 0.1586 | 0.1836 | 0.0214 | 0.0565 | 0.0633 | 0.3263 |
| | Rec | 0.2131 | 0.2275 | 0.2609 | 0.1678 | 0.2226 | 0.1452 | 0.1709 | 0.2027 | 0.1450 | 0.1093 | 0.0220 | 0.0463 | 0.0539 | 0.2121 |
| Dataset 2 | ACC | 0.5141 | 0.4952 | 0.4378 | 0.4781 | 0.4605 | 0.5002 | 0.4997 | 0.3459 | 0.3968 | 0.4359 | 0.2742 | 0.3460 | 0.3581 | 0.5803 |
| | AUPR | 0.4993 | 0.4774 | 0.4653 | 0.4441 | 0.4109 | 0.4800 | 0.4444 | 0.2760 | 0.3146 | 0.3822 | 0.2451 | 0.2932 | 0.3035 | 0.5666 |
| | AUC | 0.9456 | 0.9342 | 0.9442 | 0.9272 | 0.8822 | 0.9701 | 0.8933 | 0.9041 | 0.8400 | 0.8963 | 0.9314 | 0.7548 | 0.9463 | 0.9567 |
| | F1 | 0.2745 | 0.2841 | 0.3680 | 0.2644 | 0.2612 | 0.1916 | 0.2947 | 0.1472 | 0.1916 | 0.1882 | 0.0092 | 0.0960 | 0.1168 | 0.3065 |
| | Pre | 0.3196 | 0.3108 | 0.3625 | 0.3039 | 0.2848 | 0.3568 | 0.3395 | 0.1386 | 0.2129 | 0.2586 | 0.0475 | 0.1161 | 0.1720 | 0.3693 |
| | Rec | 0.2630 | 0.2876 | 0.4064 | 0.2469 | 0.2571 | 0.1526 | 0.2814 | 0.1688 | 0.1831 | 0.1639 | 0.0154 | 0.0873 | 0.0974 | 0.2962 |
| Win/Tie/Loss | | 11/0/1 | 10/0/2 | 9/0/3 | 12/0/0 | 10/0/2 | 10/0/2 | 12/0/0 | 12/0/0 | 12/0/0 | 11/0/1 | 12/0/0 | 12/0/0 | 11/0/1 | **144/0/12** |
| Statistic $p$-value | | 0.0005 | 0.0017 | 0.2058 | 0.0002 | 0.0012 | 0.0024 | 0.0002 | 0.0002 | 0.0002 | 0.0005 | 0.0002 | 0.0002 | 0.0005 | - |
| F-rank | | 4.17 | 4.50 | 3.25 | 6.75 | 6.92 | 6.29 | 6.17 | 9.83 | 10.13 | 8.92 | 12.50 | 13.17 | 10.42 | **2** |

Table 3: The comparison results between DHENN and its competitors in task 3, where all the drugs are unknown in DDIs graphs during training

Table 1 clearly shows that the proposed DHENN model outperforms the other baseline models on all two datasets. Looking at the total number of wins, ties, and losses, DHENN achieved 125 wins, 1 tie and 6 losses. In addition, the calculated p-values for comparisons across all datasets are less than 0.05, indicating that the statistical significance level of the performance improvement of the proposed DHENN model is 0.05. Moreover, DHENN consistently achieves the lowest F-rank value on the datasets, where a lower F-rank value indicates better model performance in comparison.

#### COMPARISON BASED ON NEW DRUGS

Tasks 2 and 3 divide the drug types into five parts, with one part containing new drugs. Subsequently, we further partitioned the data of the new drugs in the DDI dataset to create a separate test set. Table 2 and Table 3 provide a performance comparison between DHENN and the baselines for tasks 2 and tasks 3.

Based on the experimental results for task 2 and task 3, it can be observed that the proposed DHENN model outperforms the other comparison models in most cases, achieving the lowest F-rank value. Specifically, on Task 2, DHENN achieved 124 wins and 8 losses; while on Task 3, it achieved 125 wins and 7 losses. Moreover, in both Task 2 and Task 3, the p-values of the DHENN model were below 0.05. These results highlight the superior performance of the DHENN model compared to the other models.

| | ACC | F1 | Pre | Rec | F-rank |
|---|---|---|---|---|---|
| P1 | 0.9372 | 0.8660 | 0.9109 | 0.8418 | **4** |
| P1+P2 | 0.9431 | 0.8856 | 0.9276 | 0.8667 | **2.75** |
| P1+P2+P3 | 0.9454 | 0.8985 | 0.9254 | 0.8883 | **2.25** |
| P1+P2+P3+P4 | 0.9458 | 0.9032 | 0.9317 | 0.8933 | **1**[*] |
| P1 | 0.9516 | 0.9156 | 0.9386 | 0.9126 | **3** |
| P1+P2 | 0.9546 | 0.9227 | 0.9416 | 0.9133 | **2** |
| P1+P2+P3 | 0.9560 | 0.9263 | 0.9506 | 0.9235 | **1**[*] |

[*] P1, P2, P3, and P4 represent chemical entities, substructures, drugs, and molecular structures in MKG, respectively.

Table 4: The effects of incorporating different drug tail entity types of MKGs in boosting the DHENN model. Two datasets are tested: Dataset 1 (up) and Dataset 2 (down).

ABLATION STUDY (RQ.2)

Ablation experiments were conducted on the two datasets to validate the three key ideas of the proposed DHENN model. The results and findings are presented as follows.

**Effect of the MKG.** To verify the impact of constructing an MKG on model performance, this study constructed the knowledge graphs by sequentially incorporating different drug tail entity types. Table 5 shows that as different drug tail entities were added, the model's performance was continuously improved. These results verify that the constructed MKG is positive to the proposed DHENN model. More results are deferred to **Appendix**.

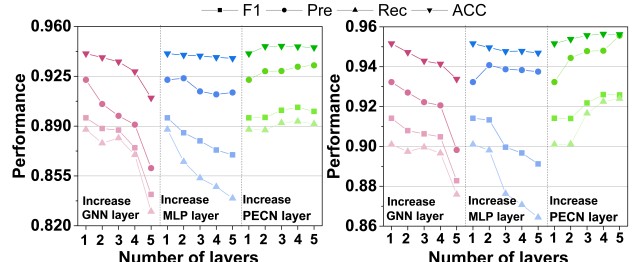

**Effect of the PECN.** To validate the effect of DHENN's deep structure, we made comparisons by increasing the number of hidden layers in the PECN, as well as increasing the number of GNN layers and MLP layers. The comparison results shown in Figure 3 indicate that as the number of hidden layers in DHENN's PECN increases, its performance consistently improves until it reaches a plateau. In contrast, increasing the number of GNN layers or MLP layers leads to a decrease in performance as the number of hidden layers increases. The results of tasks 2 and 3 are deferred to **Appendix**. These results validate that the deep structure of DHENN can enhance its performance.

Figure 3: The impact of increasing the number of GNN layer, MLP layer, and PECN layer on two datasets: left (Dataset 1) and right (Dataset 2).

**Effect of the End-to-end Structure.** The proposed DHENN model was modified to a non-end-to-end form to validate the last idea. Figure 4 shows the comparison between non-end-to-end and end-to-end DHENNs. It can be observed that the end-to-end DHENN obviously outperforms the non-end-to-end DHENN, which demonstrates that the end-to-end learning can ensure the optimal extractions and fusions of latent features from MKG.

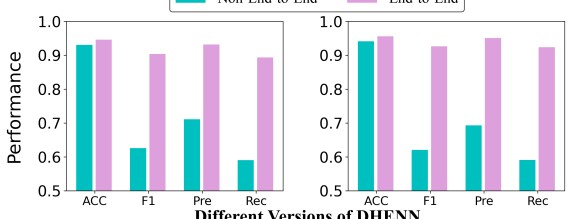

Figure 4: The impact of non-end-to-end architecture and end-to-end architecture on the performance of the DHENN model for Dataset 1 (left) and Dataset 2 (right).

HYPER-PARAMETER SENSITIVITY ANALYSIS (RQ.3)

This section identifies four crucial hyper-parameters: the dimension of drug embeddings in the drug knowledge graph ($d$), the size of the sampling neighborhood ($\mathcal{NS}$), the regularization controlling weight (RCW), and the coefficient of the cascaded loss (CCL) function in Eq.(9). To investigate the impact of these hyper-parameters, one is investigated while keeping the other fixed. The scenarios of decreasing, increasing, and staying CCL are evaluated, and found that the best performance was achieved when CCL is increasing. Please refer to the **Appendix** to see more details.

CONCLUSION

This paper proposes a novel DHENN model for accurate DDI prediction. Our model captures both binary and high-order entity relationships by constructing a multimodal knowledge graph (MKG). To enlarge the learning capacity for learning MKG representations with graph neural networks, a prediction-enhanced cascading network (PECN) is designed to dynamically incorporate shallow embeddings into deeper layers, which preserve node (drug)-level diversity extracted from the MKG construction. The MKG and PECN components are unified into an end-to-end learning framework, enabling the extraction and fusion of latent features from MKG to be optimized jointly for optimal solutions. Extensive experiments have been conducted on two real-world DDI datasets. The results demonstrate that DHENN outperforms the state-of-the-art rival models by allowing for a holistic knowledge graph embedding with deep graph representation learning in DDI prediction.

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

# APPENDIX

**Outline of the Appendix.** This Appendix serves as a supplementary file to the manuscript, providing additional insights and details. Section 1 is the explanation of adverse drug-drug interactions. Section 2 is the detailed version of the related work presented in the main manuscript. Section 3 outlines the algorithm pseudocode with time complexity analysis underlying our proposed DHENN approach, aimed at enhancing readability and reproducibility. Section 4 presents the mathematical formulas of the evaluation metrics adopted in the paper. Section 5 delves into the specifics of the baseline model. Finally, Section 6 complements the experiments conducted in the main manuscript.

# EXPLANATION OF ADVERSE DRUG-DRUG INTERACTIONS

Unknown DDIs among multiple administrated drugs in clinical settings can result in accidental toxicities and adverse reactions, some of which are, literally, deadly. Such an example is shown in Figure 5. Taking Abemaciclib alongside Bosutinib can lead to an increase in serum concentrations of Abemaciclib. Conversely, if Abemaciclib is taken at the same time as Clemastine, Abemaciclib's metabolism may be impaired,

# RELATED WORK

## DRUG FEATURE ANALYZED METHODS

The analysis of drug features plays a crucial role in predicting DDI events. In various studies, researchers assumed that similar drugs are likely to demonstrate similar DDIs. Then, they proposed approaches to acquire precise and interpretable similarity measurements by leveraging diverse types of drug features for DDI prediction (Deng et al., 2020). DeepDDI (Ryu et al., 2018) is an advanced deep-learning method designed specifically for predicting DDIs by learning drug pairs and drug-food constituent pairs. Lee et al. (Lee et al.,

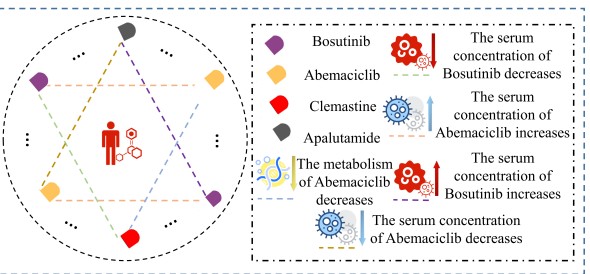

Figure 5: Examples of adverse drug-drug interaction.

2019) trained a deep feed-forward network to predict DDIs based on structural similarity profiles, Gene Ontology term similarity profiles, and target gene similarity profiles of known drug pairs. MDF-SA-DDI (Lin et al., 2022b) proposed a novel DDI events prediction model that combines multi-source drug fusion and feature fusion, while also employing transformer self-attention for offline drug feature learning. ML-RDA (Chu et al., 2019) is an advanced approach that effectively utilizes multiple drug features by incorporating the innovative unsupervised disentangling loss, CuXCov, aiming to capture diverse and informative drug characteristics. DeSIDE-DDI (Kim & Nam, 2022) is a deep learning-based framework that interprets the underlying genes in DDIs analysis, aiming to uncover the genetic factors contributing to DDIs to enhance the understanding of drug interactions. A recent multi-type DDI prediction model, MDDI-SCL (Lin et al., 2022a), was proposed by leveraging supervised contrastive learning and three-level loss functions to address various types of DDI prediction tasks proficiently.

**Discussion.** However, these methods that analyze drug features tend to prioritize acquiring extensive attributes and features of drugs, often overlooking high-order topological information and semantic relationships among drugs, targets, enzymes, transporters, molecular structures, and more. In addition, they usually employ so-called deep learning frameworks that are actually shallow to learn the underlying representations of drugs. Different from them, the proposed DHENN model is a deeper hybrid end-to-end learning framework that can extract the deep high-order topological information and semantic relationships associated with DDI events prediction.

Graph Learning-Based Methods

**Graph embedding-based.** In the realm of DDI prediction, a wide array of graph embedding methods have been employed to extract effective network-based features. These methods can be categorized into three distinct groups. Firstly, some models employ matrix decomposition techniques, utilizing the adjacency matrix as input to learn latent embeddings (Shi et al., 2019). Secondly, another category focuses on generating node sequences through random walks and subsequently learning node representations based on these sequences (Ribeiro et al., 2017). Lastly, diverse neural architectures and graph data are utilized in the final category, enabling the capture of topological connectivity patterns and leveraging the wealth of information present in drug networks (Tang et al., 2015; Wang et al., 2016).

**Knowledge graph-based.** Knowledge graphs have a profound impact in various domains, such as relation inference and recommendation systems (Wang et al., 2019). Notably, several notable techniques have emerged in DDI prediction. KGNN (Lin et al., 2020) successfully integrated graph convolutional networks with neighborhood sampling, effectively extracting valuable neighborhood relations. The AAEs (Dai et al., 2020) framework is a knowledge graph embedding approach that utilizes adversarial autoencoders, along with Wasserstein distances and GumbelSoftmax relaxation, to enhance the learning process. SumGNN (Yu et al., 2021) introduced a graph summarization module designed for subgraphs, allowing the extraction of meaningful pathways that can be easily managed and analyzed. In a similar vein, LaGAT (Hong et al., 2022) proposed a link-aware graph attention method that generates multiple attention pathways for drug entities based on the diverse links between drug pairs. Expanding on these advancements, DDKG(Su et al., 2022) takes the concept further by learning drug embeddings from their attributes within the knowledge graphs. Furthermore, DDKG incorporates neighboring node embeddings and triple facts simultaneously, leveraging an attention mechanism to capture the intricate relationships within the graphs. EmerGNN (Zhang & Yao, 2023) predicts interactions for emerging drugs by leveraging the rich information in biomedical networks. MKG-FENN (Wu et al., 2024) adopts a comprehensive and end-to-end framework to achieve optimal feature extraction and fusion. KnowDDI (Wang & Yang, 2024) enhances drug representations by adaptively leveraging rich neighborhood information from large biomedical knowledge graphs.

**Molecular graph-based.** This category of methods encompasses the prediction of molecular properties (Wang et al., 2022) and molecular interactions (Li et al., 2022). The MRGNN (Xu et al., 2019) introduced a novel approach that employs multiple graph convolution layers to extract node features from diverse neighboring nodes within a structured entity graph. MFFGNN (He et al., 2022) integrates the topological structure within molecular graphs with the interaction relationship between drugs, along with the local chemical context encoded in SMILES sequences. By combining these multiple sources of information, MFFGNN enhances the predictive performance for various molecular tasks. EPGCN-DS (Sun et al., 2020) adopts a framework based on graph convolutional networks for type-specific DDI identification from molecular structures. Additionally, Molormer (Zhang et al., 2022) leverages the two-dimensional structures of drugs as input and utilizes a lightweight attention mechanism to encode the spatial information of the molecular graph.

**Discussion.** Note that although these graph learning-based methods have delved into the topological structure and semantic relationships of DDI events, they commonly separately consider the drugs-centered direct binary relationships while ignoring the high-order information linked by drugs. In comparison, the proposed DHENN model comprehensively exploits the high-order information and mechanisms from various drugs, chemical entities, and molecular structures in one topology of MKG.

Hybrid Modeling Methods

Hybrid modeling has proven to be more effective than individual models for drug-related tasks (Chen et al., 2021). For example, the MDNN (Lyu et al., 2021) framework combines a drug knowledge graph pathway with a heterogeneous features pathway to predict DDI events. MIRACLE (Wang et al., 2021) is a novel unsupervised contrastive learning method that treats a DDI network as a multi-view graph, with each node representing a drug molecular graph instance. Deepika & Geetha (2018) employ a semi-supervised learning framework that incorporates network representation learning and

meta-learning techniques. GoGNN (Wang et al., 2020) utilizes a dual-attention mechanism to extract hierarchical features from structured entity graphs and DDI networks, enabling comprehensive information capture. Chen et al. (2021) introduced MUFFIN, a multi-scale feature fusion deep-learning model that combines drug structure and a biomedical knowledge graph for improving drug representation learning. MRCGNN (Xiong et al., 2023) integrates the features of DDI events and drug molecular graphs by GNNs.

**Discussion.** However, these hybrid modeling methods are non-end-to-end learning frameworks and may yield sub-optimal feature extractions and fusions for DDI events prediction. In comparison, the proposed DHENN model designs an end-to-end learning framework. This framework ensures that the feature extractions and fusions of DDI events are always comprehensive and optimal by seamlessly integrating the extracted features throughout the learning process.

## ALGORITHM DESIGN AND TIME COMPLEXITY ANALYSIS

By analyzing the proposed DHENN model, its algorithm is designed in **Algorithm 1**.

First, we construct the DDI matrix $\mathcal{Y}$ and the multi-modal knowledge graph $\mathcal{G}$. Then we initialize the multi-modal knowledge graph $\mathcal{G}$. In steps 4-5, we randomly sample a fixed-size sample $\{\mathcal{N}_s\}_{l=1}^{L}$ from the drug knowledge graph, where $\mathcal{N}_l$ represents the neighborhood size at the $l$-th layer of GNN and $L$ represents the number of layers in the GNN. In steps 6-12, We employed GNN to compute the higher-order structure and semantic relationships among drugs, and concatenated the representations of drug pairs. In steps 14-19, We employ a cascaded deep structure to predict drug representations to enhance predictive performance.

From the overall algorithmic perspective, DHENN is divided into two parts: GNN extracts drug representations, and the cascaded deep structure predicts DDI events. The time complexity of the GNN part is $O(N_{DDI} \times D \times \mathcal{N}_s \times L)$, where $N_{DDI}$ represents the number of DDIs, and $D$ represents the dimension of drug encodings. In the cascaded deep structure part, the corresponding time complexity is $O(N_{DDI} \times N \times (D + C))$, where $N$ represents the number of cascaded layers and $C$ represents the number of DDI prediction categories. Therefore, the overall time complexity of the final model is $O(N_{DDI} \times ((D \times \mathcal{N}_l \times L) + (N \times (D + C))))$.

---

**Algorithm 1:** DHENN Algorithm

> **input** : DDI matrix $\mathcal{Y}$, multi-modal knowledge graph $\mathcal{G}$.
> **output:** $\Gamma(d_i, d_j | \mathcal{Y}, \mathcal{G})$
>
> **1** *Initialization $\mathcal{G}$;*
> **2** **while** *not converge* **do**
> **3**    **for** $(d_i, d_j)$ *in* $\mathcal{Y}$ **do**
> **4**      $\{\mathcal{N}_l\}_{l=1}^{L} \leftarrow$ *Neighborhood Sampling(entity e)*;
> **5**      $e^0 \leftarrow e, \forall e \in \mathcal{N}_0$;
> **6**      **for** $l = 1, ..., L$ **do**
> **7**        **for** $e \in \mathcal{N}_l$ **do**
> **8**          $e_{\mathcal{N}_l}^{(l)} \leftarrow \sum_{t_n \in \mathcal{N}_l(e)} \pi_{(e,r_{in})}^{(l)} e_{t_n}^{(l-1)}$;
> **9**        **end**
> **10**      **end**
> **11**      $E_{d_i}^{j+1} \leftarrow e_{d_i}^{(l)}, E_{d_j}^{j+1} \leftarrow e_{d_j}^{(l)}$;
> **12**      $\hat{E}_{d_{i,j}} \leftarrow \hat{E}_{d_j} \oplus \hat{E}_{d_i}$;
> **13**      $y_{ij} \leftarrow 0$;
> **14**      **for** $n = 1, ..., N$ **do**
> **15**        $\hat{E}_{d_{i,j}} \leftarrow \hat{E}_{d_{i,j}} \oplus y_{ij}$;
> **16**        *Calculate* $y_{ij} = f\left(\hat{E}_{d_{i,j}}\right)$;
> **17**        *Update parameters* $\Theta$;
> **18**      **end**
> **19**    **end**
> **20** **end**

---

## EVALUATION METRIC

Regarding the evaluation metrics for model assessment, we utilize a diverse array of multi-class classification evaluation metrics to ensure a comprehensive understanding of the model's performance. These metrics include accuracy (ACC), area under the precision-recall curve (AUPR), area under the receiver operating characteristic curve (AUC), F1 score, precision, and recall (Deng et al., 2020). The formulas for these metrics are as follows:

$$\text{ACC} = \frac{\sum_{i=1}^{n} TP_i}{\sum_{i=1}^{n} TP_i + \sum_{i=1}^{n} FN_i} \tag{10}$$

$$\text{Precision} = \left( \sum_{i=1}^{n} \frac{TP_i}{TP_i + FP_i} \right) / n \tag{11}$$

$$\text{F1} = 2 * \frac{\Pr ecision * \text{Recall}}{\Pr ecision + \text{Recall}} \tag{12}$$

$TP_i$ denotes the situation where both the actual disease and the predicted disease are the $i$-th type. Conversely, $FN_i$ signifies a scenario where the actual disease is the $i$-th type, but the prediction erroneously indicates a different disease. On the other hand, $FP_i$ occurs when the actual disease differs from the $i$-th type, yet the prediction incorrectly identifies it as the $i$-th disease. Lastly, $TN_i$ represents a correct prediction where the actual disease is not the $i$-th type, and the prediction accurately reflects this. It is worth noting that $n$ represents the types of events that will occur.

$$\text{TPR} = \text{Re}call = \left( \sum_{i=1}^{n} \frac{TP_i}{TP_i + FN_i} \right) / n \tag{13}$$

$$\text{FPR} = \left( \sum_{i=1}^{n} \frac{FP_i}{FP_i + TN_i} \right) / n \tag{14}$$

When plotting the False Positive Rate ($FPR$) on the x-axis and the True Positive Rate ($TPR$) on the y-axis, the AUC (Area Under the Curve) represents the total area enclosed by the $FPR$-$TPR$ curve. Conversely, when using $\text{Re}call$ as the x-axis and $Precision$ as the y-axis, the AUPR (Area Under the $Precision$-$\text{Re}call$ Curve) denotes the enclosed area beneath the $Precision$-$\text{Re}call$ curve.

## BASELINE MODEL

Owing to the extensive nature of the text, we shall focus on presenting an overview of the baseline model in this context. Specifically, we will introduce nine cutting-edge models: MKG-FENN (Wu et al., 2024), KnowDDI (Wang & Yang, 2024), EmerGNN (Zhang & Yao, 2023), MDDI-SCL (Lin et al., 2022a), MDF-SA-DDI (Lin et al., 2022b), MDNN (Lyu et al., 2021), DDIMDL (Deng et al., 2020), Lee et al.'s methods (Lee et al., 2019), and DeepDDI (Ryu et al., 2018). Additionally, we will also consider several traditional classification methods, namely DNN, RF, KNN, and LR (Deng et al., 2020), for comparison. A comprehensive breakdown of the comparison models is detailed in Table 6.

|             | ACC    | F1     | Pre    | Rec    | F-rank  |
|-------------|--------|--------|--------|--------|---------|
| P1          | 0.9372 | 0.8660 | 0.9109 | 0.8418 | **14.75** |
| P2          | 0.9381 | 0.8796 | 0.9114 | 0.8705 | **12.00** |
| P3          | 0.9373 | 0.8811 | 0.9111 | 0.8725 | **12.00** |
| P4          | 0.9374 | 0.8780 | 0.9001 | 0.8642 | **14.00** |
| P1+P2       | 0.9431 | 0.8856 | 0.9276 | 0.8667 | **7.25**  |
| P1+P3       | 0.9406 | 0.8901 | 0.9230 | 0.8777 | **7.50**  |
| P1+P4       | 0.9418 | 0.8931 | 0.9195 | 0.8805 | **6.63**  |
| P2+P3       | 0.9418 | 0.8857 | 0.9219 | 0.8713 | **8.63**  |
| P2+P4       | 0.9420 | 0.8882 | 0.9263 | 0.8730 | **6.63**  |
| P3+P4       | 0.9415 | 0.8823 | 0.9170 | 0.8681 | **10.88** |
| P1+P2+P3    | 0.9454 | 0.8985 | 0.9254 | 0.8883 | **2.75**  |
| P1+P2+P4    | 0.9427 | 0.8937 | 0.9305 | 0.8868 | **3.50**  |
| P1+P3+P4    | 0.9429 | 0.8958 | 0.9170 | 0.8828 | **5.38**  |
| P2+P3+p4    | 0.9420 | 0.8910 | 0.9176 | 0.8753 | **7.13**  |
| P1+P2+P3+P4 | 0.9458 | 0.9032 | 0.9317 | 0.8933 | **1**[*]  |
| P1          | 0.9516 | 0.9156 | 0.9386 | 0.9126 | **5.50**  |
| P2          | 0.9494 | 0.9178 | 0.9370 | 0.9194 | **4.50**  |
| P3          | 0.9491 | 0.9102 | 0.9354 | 0.9109 | **7.00**  |
| P1+P2       | 0.9546 | 0.9227 | 0.9416 | 0.9133 | **3.25**  |
| P1+P3       | 0.9535 | 0.9176 | 0.9418 | 0.9150 | **3.75**  |
| P2+P3       | 0.9531 | 0.9217 | 0.9423 | 0.9177 | **3**     |
| P1+P2+P3    | 0.9560 | 0.9263 | 0.9506 | 0.9235 | **1**[*]  |

[*] P1, P2, P3, and P4 represent chemical entities, sub-structures, drugs, and molecular structures in MKG, respectively.

Table 5: The impact of combining different sections of the MKG on the performance of the DHENN model for two datasets: up (Dataset 1) and down (Dataset 2).

| Model | Description |
|---|---|
| MKG-FENN (Wu et al., 2024) | It is a knowledge graph-based method that adopts acomprehensive and end-to-end framework to achieve optimal feature extraction and fusion, *AAAI 2024*. |
| KnowDDI (Wang & Yang, 2024) | It is a knowledge graph-based method that utilizes rich neighborhood information from large biomedical knowledge graphs to enhance drug representations, *Communications Medicine 2024*. |
| EmerGNN (Zhang & Yao, 2023) | It is a knowledge graph-based method that predicts interactions for emerging drugs by leveraging rich information from biomedical networks, *Nature Computational Science 2023*. |
| MDDI-SCL (Lin et al., 2022a) | It is a method drug based on features analysis that leverages supervised contrastive learning as its foundation, *Journal of Cheminformatics 2022*. |
| MDF-SA-DDI (Lin et al., 2022b) | It is a method based on drug feature analysis that adopts multi-source drug fusion, incorporating multi-source features and the transformer self-attention mechanism, *Briefings in Bioinformatics 2022*. |
| MDNN (Lyu et al., 2021) | It is a hybrid method that combines a drug knowledge graph pathway and a heterogeneous features pathway to predict drug-drug interaction events, *IJCAI 2021*. |
| DDIMDL (Deng et al., 2020) | It is a method based on drug feature analysis that combines multiple drug profiles using deep learning techniques, *Bioinformatics 2020*. |
| Lee et al.'s methods (Lee et al., 2019) | It is a method based on drug feature analysis that adopts a novel deep learning model aimed at enhancing classification accuracy, *BMC Bioinform 2019*. |
| DeepDDI (Ryu et al., 2018) | It is a representative matrix factorization model that decomposes the user-item matrix data for use in recommender systems, *Proc. Natl. Acad. Sci. U.S.A. 2018*. |
| DNN (Deng et al., 2020) | It is a traditional classification method deep neural network. |
| RF (Deng et al., 2020) | It is a traditional classification method random forest. |
| KNN (Deng et al., 2020) | It is a traditional classification method k-nearest neighbour. |
| LR (Deng et al., 2020) | It is a traditional classification method logistic regression. |
| DHENN | Our model is a multimodal, deep learning-based predictive system with a cascade structure for accurate predictions. |

Table 6: Descriptions of all the contrasting models.

ABLATION STUDY

**Effect of the MKG.** To evaluate the impact of constructing an MKG on model performance, this study sequentially incorporated different drug tail entity types into the knowledge graph construction. As shown in Table 5, the model's performance was continuously improved as different drug tail entities were added to the MKG. These results verify that the constructed MKG is beneficial for the proposed DHENN model.

**Effect of the PECN.** Table 7 shows the performance comparison between two versions of DHENN with One Layer and Deeper Layers of PECN, respectively, on three tasks. The percentages of performance improvement by deeper layers of PECN range from 0.56% to 18.03%. These results validate that the deep structure of PECN can enhance the performance of DHENN.

| Dataset | Version of DHENN | Task 1 | | | | Task 2 | | | | Task 3 | | | |
|---------|------------------|--------|-----|-----|-----|--------|-----|-----|-----|--------|-----|-----|-----|
| | | F1 | Pre | Rec | ACC | F1 | Pre | Rec | ACC | F1 | Pre | Rec | ACC |
| Dataset 1 | One layer of PECN | 0.8958 | 0.9225 | 0.8876 | 0.9409 | 0.5185 | 0.6128 | 0.4965 | 0.6803 | 0.2086 | 0.2764 | 0.2120 | 0.4668 |
| | Deeper layers of PECN | 0.9032 | 0.9327 | 0.8933 | 0.9462 | 0.5362 | 0.6413 | 0.5187 | 0.6910 | 0.2252 | 0.3263 | 0.2159 | 0.4847 |
| | Improvement Percentage | **0.83%** | **1.11%** | **0.64%** | **0.56%** | **3.41%** | **4.66%** | **4.46%** | **1.57%** | **7.95%** | **18.03%** | **1.83%** | **3.82%** |
| Dataset 2 | One layer of PECN | 0.9141 | 0.9323 | 0.9010 | 0.9516 | 0.5745 | 0.7026 | 0.5289 | 0.7246 | 0.2723 | 0.3482 | 0.2618 | 0.5578 |
| | Deeper layers of PECN | 0.9260 | 0.9556 | 0.9240 | 0.9562 | 0.5882 | 0.7308 | 0.5488 | 0.7373 | 0.3065 | 0.3674 | 0.2938 | 0.5802 |
| | Improvement Percentage | **1.30%** | **2.50%** | **2.55%** | **0.48%** | **2.38%** | **4.01%** | **3.77%** | **1.75%** | **12.57%** | **5.50%** | **12.21%** | **4.01%** |

Table 7: Performance comparison between two versions of DHENN with different layers of PECN.

HYPER-PARAMETER SENSITIVITY ANALYSIS

In this study, we have identified four pivotal parameters: the dimensionality of drug embeddings within the drug knowledge graph ($d$), the extent of the sampling neighborhood ($\mathcal{N}S$), the regularization controlling weight (RCW), and the coefficient of the cascaded loss function (CCL). Hyper-parameter sensitivity experiments are presented in Figure 6.

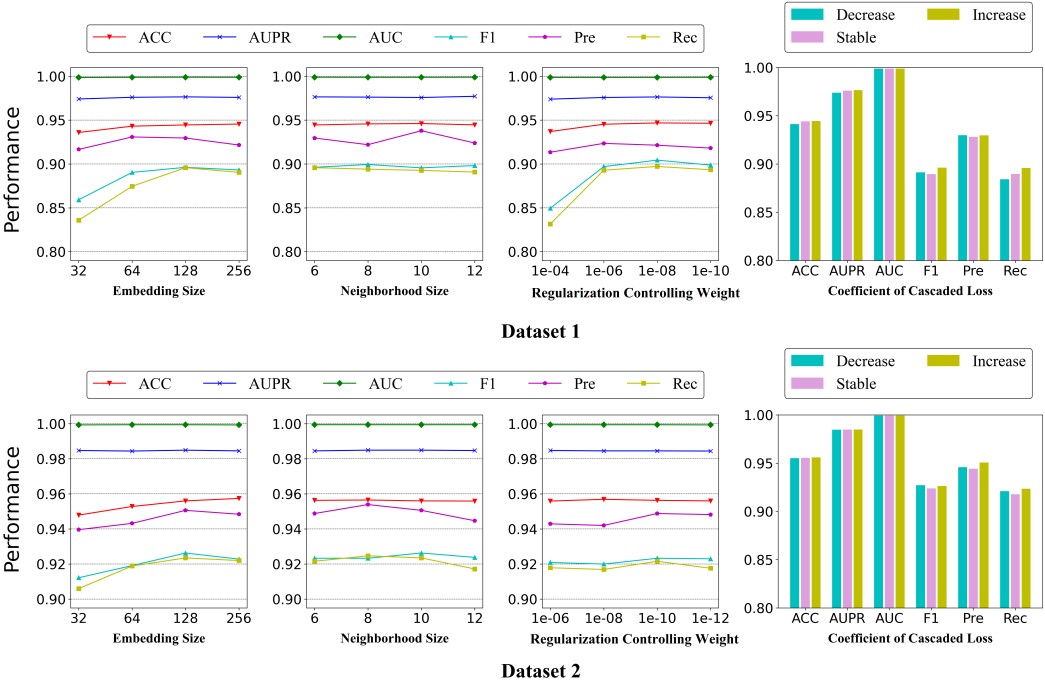

Figure 6: Sensitivity analysis of parameters in two datasets for exploring the impact of parameter variations on results.

**Effect of embedding dimension.** The performance of the model can be affected by changing the embedding dimensions, and we investigated the influence of varying the value of $d$ on model performance. Choosing an appropriate value for $d$ enables the model to capture a sufficient amount of drug and entity information, resulting in improved performance. From Figure 6, we can see that Dataset 1 utilized an embedding dimension of $d = 128$, and Dataset 2 also employed $d = 128$.

**Effect of neighborhood size.** We examined how the performance of the model is affected by varying the size of the sampled neighborhood. Figure 6 demonstrates the optimal values of the neighborhood sample ($\mathcal{NS}$) for the two datasets. In Dataset 1, the optimal $\mathcal{NS}$ value is 10. Similarly, in Dataset 2, the optimal $\mathcal{NS}$ value is also 10. When the neighborhood size was too small, the model faced difficulties in effectively organizing the information. On the other hand, when $\mathcal{NS}$ was too large, the model became more susceptible to being influenced by noise.

**Effect of regularization controlling weight.** The impact RCW on the model's performance is substantial. After conducting several experiments, we have determined that fine-tuning the RCW can significantly improve the model's performance. Figure 6 reveals that Dataset 1 achieved the best model performance with an optimal RCW value of 1e-8, and Dataset 2 had the optimal RCW value of 1e-10.

**Effect of coefficient of cascaded loss.** Figure 6 discusses the impact of CCL on the model. By studying the performance of the model with varying CCL values across two datasets, it can observe that the model achieved the best results when the CCL was in an increasing state.

