# OpenReview forum: "DHENN: A Deeper Hybrid End-to-end Neural Network for Highly Accurate Drug-Drug Interaction Events Prediction"
_ICLR.cc/2025/Conference — Submitted to ICLR 2025_

### Official Review · Reviewer_FpBb · 2024-10-30

**Soundness:** 2
**Presentation:** 2
**Contribution:** 2
**Rating:** 3
**Confidence:** 5

**Summary:**

This paper proposes an end-to-end DDI prediction model to enhance the high-order interactions between biomedical entities and the learning capacity of adopted neural networks. The DDI prediction task is interesting. However, the motivations are convincing enough. The experimental results look positive, but the key KG-based prediction methods are missing.

**Strengths:**

The paper is well-structured.

**Weaknesses:**

Novelty:
The novelty is limited and this is an incremental work based on MKG-FENN [1]. The authors should clarify the key differences between them.

Motivation:
This paper proposes three motivations for the traditional DDI methods' failing short in the high-order interactions between biomedical entities, the limited learning capacity, and the decoupled learning framework. For the first motivation, this paper considers the relational semantics with topological structures as high-order interactions, however, some works have considered this information[2-5]. For the second motivation, I suggest that the authors elaborate on what the advantages of deep neural networks are for DDI prediction, and from the experimental results, there is no huge advantage of using deep neural networks over the previous methods. For the third motivation, these works [2-6] are all end-to-end methods for DDI prediction. I suggest the authors provide more evidence or analysis to demonstrate their strengths.

Experiment:
For the baseline methods, many KG-based works [1-5] are missing. So the experimental results are not convincing. The authors can further discuss these methods and explain how your approach differs from or improves upon these methods.

Some minor suggestions, the authors can provide more discussion on these.
1. In the Method section, Equation (5) lacks sufficient explanation of the symbol, as the symbol σ is used without specifying the type of activation function it represents.
2. The example "986" is used as a SMILES encoding for a drug, but it is not a typical SMILES sequence. A more representative SMILES sequence is recommended.
3. The Related Work section mentions four categories of methods, but the main experiment’s baseline selection does not include models from each of these four categories.
4. The main experiment analysis is insufficient, only presenting results without in-depth discussion on the reasons for performance improvements or why it outperforms certain types of methods. There is also no analysis of the strengths and weaknesses of models other than the proposed one. The authors can provide more analysis.
5. The descriptions of Tasks 2 and 3 in the experiment section are not detailed enough, and the overview and differences between the two tasks are unclear.
6. Table 4 lacks an explanation, and there is no analysis of the experimental results for different combinations of the four entity types.

[1] Wu, D., Sun, W., He, Y., Chen, Z., & Luo, X. (2024, March). MKG-FENN: A Multimodal Knowledge Graph Fused End-to-End Neural Network for Accurate Drug–Drug Interaction Prediction. In Proceedings of the AAAI Conference on Artificial Intelligence (Vol. 38, No. 9, pp. 10216-10224).

[2] Chen, Y., Ma, T., Yang, X., Wang, J., Song, B., & Zeng, X. (2021). MUFFIN: multi-scale feature fusion for drug–drug interaction prediction. Bioinformatics, 37(17), 2651-2658.

[3] Yu, Y., Huang, K., Zhang, C., Glass, L. M., Sun, J., & Xiao, C. (2021). SumGNN: multi-typed drug interaction prediction via efficient knowledge graph summarization. Bioinformatics, 37(18), 2988-2995.

[4] Wang, Y., Yang, Z., & Yao, Q. (2024). Accurate and interpretable drug-drug interaction prediction enabled by knowledge subgraph learning. Communications Medicine, 4(1), 59.

[5] Zhang, Y., Yao, Q., Yue, L., Wu, X., Zhang, Z., Lin, Z., & Zheng, Y. (2023). Emerging drug interaction prediction enabled by a flow-based graph neural network with biomedical network. Nature Computational Science, 3(12), 1023-1033.

[6] Lin, X., Quan, Z., Wang, Z. J., Ma, T., & Zeng, X. (2020, July). KGNN: Knowledge Graph Neural Network for Drug-Drug Interaction Prediction. In IJCAI (Vol. 380, pp. 2739-2745).

**Questions:**

1. In the Method section, it is noted that MKG includes various entities and relationships. Does the model have a mechanism to adaptively assign different weights to these relationships?
2. Can the model handle cases where data is incomplete or certain modality information is missing?
3. The paper mentions using a fixed neighborhood size for sampling. Has dynamic neighborhood sampling been considered, where sampling size could be adjusted based on node importance or graph structure? Would this improve model performance on DDI networks with varying structural complexities?

---

> ### Author Response · Authors · 2024-11-25
>
> We thank the reviewer for the thoughtful and detailed feedback. Below, we address the weaknesses and questions one by one.
>
> +++ Regarding **Weaknesses** +++
>
> **W1**: Limited novelty and incremental improvement over [1].
>
> **A1**: We respectfully clarify that DHENN represents a significant departure from MKG-FENN [1]. While MKG-FENN employs shallow neural networks to avoid oversmoothing, our DHENN introduces a novel PECN architecture design, which allows for deeper neural networks to improve the learning efficacy without bottlenecked by oversmoothing. This allows DHENN to capture richer, higher-order relationships in the data, which are critical for DDI prediction. The results in Tables 1–3 clearly demonstrate that DHENN significantly outperforms MKG-FENN across all tasks, establishing its superiority in terms of predictive accuracy and robustness.
>
> **W2**: Lack of convincing evidence for proposed motivations.
>
> **A2**: We elaborate our three motivators. 1) For high-order interactions, we argue that our MKG is more comprehensive than the referenced methods [2–5] by unifying four types of drug-centered binary relationships: <drugs, chemical entities>, <drugs, substructures> (SMILES), <drugs, drugs> (DDIs), and <drugs, molecular fingerprints> (MACCS), all into a single MKG topology. This comprehensive integration allows DHENN to better explore complex relational semantics and topological structures, which are underrepresented in prior works.
>
> 2) For learning capacity, our PECN demonstrates a clear advantage by enabling deeper NNs without oversmoothing, as evidenced by Figure 3. Unlike [1–6], which rely on shallow (one-layer) network due to oversmoothing concerns, DHENN achieves improved performance through a deeper PECN, bridging the gap between learning capacity and predictive performance. 3) For end-to-end learning, while we acknowledge that [2–6] are also end-to-end methods, they often rely on less comprehensive knowledge graphs, limiting their ability to capture diverse drug-centered relationships. DHENN integrates a richer MKG with four different drug-centric information and leverages PECN to learn it, significantly outperforming these methods as demonstrated in our experiments.
>
> **W3**: Missing comparisons with key KG-based methods [1–5].
>
> **A3**: We appreciate the suggestion for additional comparisons. However, we would like to note that, in the original manuscript, we have already compared DHENN with MKG-FENN [1].  To complete the state-of-the-art, we have now included comparative experiments with KnowDDI (2024) [4] and EmerGNN (2023) [5] in the revised manuscript. We do not include [2–3] because they were published in 2021, and have been outperformed by the latest models, such as  [1, 4, 5] published in 2023 or 2024. The new results confirm that DHENN consistently outperforms all thirteen baseline methods, including [1, 4, 5].
>
> **W4**: A more representative SMILES sequence is recommended.
>
> **A4**: The example “986”  is a numerical attribute derived from the molecular structure of Lovastatin. Please refer to the A3 response to Reviewer nZSN for more details. For better clarity, we replace “986” with a more representative SMILES sequence and provide additional explanations in the revised manuscript.
>
>
> **W5**: Descriptions of Tasks 2 and 3 are unclear.
>
> **A5**:  In the revised manuscript, we have clarified their differences. Task 2 evaluates DHENN’s ability to predict DDIs when partial drugs are excluded from the training set (partial generalization), while Task 3 assesses its performance when all drugs are excluded (complete generalization). These tasks reflect progressively more challenging scenarios, highlighting DHENN’s robustness in generalization.
>
> +++ Regarding **Questions** +++
>
> **Q1**: Does the model adaptively assign different weights to relationships in the MKG?
>
> **A1**: Yes, the relationships in the MKG are adaptively fused through our end-to-end learning approach. The model learns to assign weights implicitly based on the contributions of different relationships to the prediction task, ensuring that critical relationships are prioritized.
>
>
> **Q2**: Can the model handle incomplete data or missing modality information?
>
> **A2**: Yes, DHENN can handle incomplete data or missing modalities. Its modular architecture ensures that predictions remain robust even when certain types of information are unavailable. This capability makes it particularly suitable for real-world applications where data completeness is often a challenge.
>
>
> **Q3**: Has dynamic neighborhood sampling been considered?
>
> **A3**:  We investigated the impact of neighborhood sampling in Figure 6 (Appendix). Our analysis showed that a small size of neighbors hinder information propagation, while excessively large neighborhoods introduce noise. While we currently adopt a fixed neighborhood size to balance these factors, we acknowledge the potential benefits of dynamic sampling and will explore this direction in future work.

---

> > ### Comment · Reviewer_FpBb · 2024-11-26
> >
> > Thank you for the authors' response. The experimental results suggest that increasing the depth of PECN does not result in significant improvements. Additionally, oversmoothing does not appear to be a major concern for DDI prediction, as comparable performance can be achieved with shallow networks. I would maintain my score.

---

> ### Author Response · Authors · 2024-11-30
>
> We thank the reviewer for the discussions and providing us the opportunity to further clarify your concerns. We address them point by point as follows.
>
> **Q1**: The experimental results suggest that increasing the depth of PECN does not result in significant improvements. Additionally, oversmoothing does not appear to be a major concern for DDI prediction, as comparable performance can be achieved with shallow networks.
>
> **A1**: Oversmoothing is one severe limitation of GNN [7-8], which makes the indistinguishable representations of nodes in different classes. Notably, as demonstrated in [1, 9], oversmoothing also causes the indistinguishable representations of drugs by GNNs, which limits the performance of DDI prediction. Thus, **we argue that oversmoothing is a major concern for DDI prediction**.  To aid this issue, we design deeper PECN to mitigate the oversmoothing associated with the nodal feature extraction in DDI prediction. **Table 7 (Appendix) clearly shows that our deeper PECN can significantly improve the performance, such as by 18.03% at most**. Besides, Tables 1-3 (manuscript) clearly show that our model significantly outperforms nine state-of-the-art models.
>
> **Q2**: Regarding previous comments that  ”these works [2-6] are all end-to-end methods for DDI prediction”.
>
> **A2**: We clarify that non-end-to-end is not the major drawback of previous studies [2-6]. **Their main drawback is to consider drug-centered KGs partially or not unify multiple KGs into an end-to-end learning way in both embedding and prediction stages**. For example, MUFFIN [2] adopts the pretrained-GNN to obtain the embeddings of molecular graphs of drugs, which is not included in the remaining end-to-end learning process. Another example, the classifier parameter “W_c” of KnowDDI [4] is not included in its loss function, which means that the prediction stage is not included in the end-to-end learning process.
>
>
> [1] Wu, D., Sun, W., He, Y., Chen, Z., & Luo, X. (2024, March). MKG-FENN: A Multimodal Knowledge Graph Fused End-to-End Neural Network for Accurate Drug–Drug Interaction Prediction. In Proceedings of the AAAI Conference on Artificial Intelligence (Vol. 38, No. 9, pp. 10216-10224).
>
> [2] Chen, Y., Ma, T., Yang, X., Wang, J., Song, B., & Zeng, X. (2021). MUFFIN: multi-scale feature fusion for drug–drug interaction prediction. Bioinformatics, 37(17), 2651-2658.
>
> [3] Yu, Y., Huang, K., Zhang, C., Glass, L. M., Sun, J., & Xiao, C. (2021). SumGNN: multi-typed drug interaction prediction via efficient knowledge graph summarization. Bioinformatics, 37(18), 2988-2995.
>
> [4] Wang, Y., Yang, Z., & Yao, Q. (2024). Accurate and interpretable drug-drug interaction prediction enabled by knowledge subgraph learning. Communications Medicine, 4(1), 59.
>
> [5] Zhang, Y., Yao, Q., Yue, L., Wu, X., Zhang, Z., Lin, Z., & Zheng, Y. (2023). Emerging drug interaction prediction enabled by a flow-based graph neural network with biomedical network. Nature Computational Science, 3(12), 1023-1033.
>
> [6] Lin, X., Quan, Z., Wang, Z. J., Ma, T., & Zeng, X. (2020, July). KGNN: Knowledge Graph Neural Network for Drug-Drug Interaction Prediction. In IJCAI (Vol. 380, pp. 2739-2745).
>
> [7] Chen et al., Measuring and relieving the oversmoothing problem for graph neural networks from the topological view. AAAI 2020.
>
> [8] Liu et al., Towards deeper graph neural networks. In Proceedings of the 26th ACM SIGKDD international conference on knowledge discovery & data mining, pp. 338–348, 2020.
>
> [9] Tengfei Lyu, Jianliang Gao, Ling Tian, Zhao Li, Peng Zhang, and Ji Zhang. Mdnn: A multimodal deep neural network for predicting drug-drug interaction events. In IJCAI 2021

---

### Official Review · Reviewer_4sRf · 2024-10-31

**Soundness:** 3
**Presentation:** 3
**Contribution:** 2
**Rating:** 8
**Confidence:** 3

**Summary:**

This paper presents DHENN, a deep learning framework for predicting drug-drug interactions (DDIs). DHENN tries to address three main limitations current DDI suffer from:
1) Simplification of dependency structures, failing to model high-order interactions. They solve this by building a multimodal knowledge graph, from where complex interactions can be learned.
2) over-smoothing effects of neural network approaches. They solve this by implementing a cascading network that can leverage shallow embeddings and prevent over-smoothing.
3) decoupling of representation and prediction, yielding suboptimal solutions. They adress this by training both representation and prediction modules in an end-to-end manner.
The authors perform several experiments across various DDI datasets and state-of-the-art prediction models, showing improved performance in many occasions. The paper is well written and presented.

**Strengths:**

1- The manuscript is clear and well explained and the techniques behind the developed model present novelty.

2- The performed analysis is extensive, across several state-of-the-art methodologies and datasets.

3- The presented methodology proofs to over-perform, in a statistically significant manner, many state-of-the-art DDI approaches.

4- They present an extensive ablation study tackling several important concepts of the model, such as the over-smoothing effect or the multimodal graph importance, which adds robustness to the work.

**Weaknesses:**

1- The paper would benefit from an analysis of the model’s computational performance, such as as training time and memory/GPU usage.

2- It is unclear how different drugs perform based on the connectivity on the multimodal graph. Can this model be applied to new drugs that may not have connections within the multimodal graph?

3 - Although the authors present an analysis based on new drugs, the paper could benefit from a further strict evaluation on generalization capabilities. This could encompass train/test folds to be different datasets i.e. multimodal networks, with low intersection among them. Maybe network A can be drugbank and network B can be a smaller network coming from a different database.

**Questions:**

1 - Regarding the comparison based on new drugs, are the authors reporting the 5th part, i.e., the one containing the new drugs? Are these parts equally-sized? I would appreciate further clarification.

2 - How is the non-end-to-end implemented? is it a sequential two-phases training? I would appreciate if the authors can provide a detailed description of the implementation.

---

> ### Author Response · Authors · 2024-11-25
>
> We thank the reviewer for the encouraging and constructive feedback, where we address your concerns point by point below.
>
> +++ Regarding **Weaknesses** +++
>
> **W1**: Further strengthen the paper by analyzing the model’s computational performance.
>
> **A1**: Thank you for the suggestion. We have analyzed the training time and memory/GPU usage across all models. The results are as follows:
>
> | Model               | Training Time (s) | Memory Usage (GB) |
> |----------------------|-------------------|-------------------|
> | MKG-FENN            | 13,440           | 1.3               |
> | MDDI-SCL            | 21,280           | 17.1              |
> | MDF-SA-DDI          | 37,300           | 17.0              |
> | DDIMDL              | 280              | 9.1               |
> | MDNN                | 8,640            | 1.1               |
> | Lee et al.'s  | 208,000          | 10.4              |
> | DeepDDI             | 472              | 1.3               |
> | DNN                 | 278              | 11.1              |
> | RF                  | 1,204            | 8.0               |
> | KNN                 | 74               | 6.9               |
> | LR                  | 290              | 8.2               |
> | DHENN (ours.)               | 13,920           | 2.7               |
>
> We will add these results plus discussions in our camera ready.
>
> **W2**: How different drugs perform based on the connectivity in the multimodal graph. Can this model be applied to new drugs that may not have connections within the multimodal graph?
>
> **A2**: Our Multimodal Knowledge Graph (MKG) integrates four types of drug-centered relationships: <drugs, chemical entities>, <drugs, substructures> (coding by SMILES), <drugs, drugs> (i.e., DDI events), and <drugs, molecular fingerprints>(coding by MACCS). The MKG ensures connectivity through these diverse relationships. In our experiments, we create two learning setups, namely, Task 2 (partial drugs unknown) and Task 3 (all drugs unknown), with their results showing in Tables 2 and 3, respectively. Through evaluating the two setups where new drugs are introduced without prior connections in <drugs, drugs> interactions, we provide affirmative answer that our DHENN model can generalize to new drugs, even when they lack direct connections in the <drugs, drugs> interaction graph.
>
> **W3**: Stricter evaluation of generalization capabilities?
>
> **A3**:  In our current work, we employed five-fold cross-validation to evaluate generalizability on a single dataset. We acknowledge that evaluating our model by training and testing on different datasets with minimal overlap (e.g., DrugBank vs. a smaller external dataset) would provide a more rigorous assessment of its domain adaptation capabilities. This evaluation is valuable for discovering and testing new drugs. However, due to time constraints and the scope of this study, we were unable to conduct such an extensive cross-dataset evaluation. We plan to explore this direction in future work, as it aligns closely with our long-term goals of enhancing model robustness and applicability in novel drug development.
>
> +++ Regarding **Questions** +++
>
> **Q1**:  Regarding the comparison based on new drugs, are the authors reporting the 5th part, i.e., the one containing the new drugs? Are these parts equally sized?
>
> **A1**: Yes, the results reported in Tables 2-3 are for the new drugs in two datasets. This is how we did: we divided the dataset into five equal parts. In each repeated experiment, 1/5 of the drugs were excluded from the training dataset and used exclusively for testing. Namely, 20% of the drugs were new to the model in each run. We implemented cross-validation and performed checks to prevent any potential data leakage between training and test.
>
> **Q2**: How is the non-end-to-end approach implemented?
>
> **A2**: Correct. We implemented it as a sequential two-phase training process with different optimization objectives. The GNN phase optimizes the reconstruction loss of graph structures, and the PECN phase minimizes the prediction loss for DDI events.

---

> > ### Comment · Reviewer_4sRf · 2024-11-29
> >
> > I appreciate the author's response. I'm maintaining my score.

---

> > > ### Author Response · Authors · 2024-11-29
> > >
> > > We highly appreciate the reviewer's response. Your constructive comments are very helpful for improving this study.

---

### Official Review · Reviewer_tLVQ · 2024-11-03

**Soundness:** 3
**Presentation:** 3
**Contribution:** 2
**Rating:** 5
**Confidence:** 4

**Summary:**

This study presents a drug-drug interaction (DDI) prediction method, DHENN, designed to overcome the limitations of previous approaches, including the neglect of high-order information in drug knowledge graphs (KGs), oversmoothing issues, and non-end-to-end frameworks. The authors introduce a message-passing process for drug representation learning and a cascading layer called PECN for DDI prediction. DHENN's performance is evaluated against 12 baseline models across three tasks and two benchmark datasets, complemented by an ablation study and hyperparameter sensitivity analysis.

**Strengths:**

- The model architecture and computational processes are well-articulated.
- The inclusion of multiple competitive baselines offers a comprehensive assessment of DHENN’s performance and competitiveness.
- The ablation study is designed to directly address and evaluate the core limitations identified by the authors.

**Weaknesses:**

-  Regarding limitation 1 (High-order interactions): DHENN utilizes a single-layer GNN, which only captures first-order relationships for representation learning. This raises the question of **whether it is genuinely more effective at capturing high-order information compared to other models**. Moreover, referring to "high-order information" as a significant limitation seems overly generalized, as this issue could be easily addressed by stacking multiple GNN layers to propagate higher-order interactions throughout the graph.
-  Regarding limitation 2 (Oversmoothing): Given that DHENN only employs a one-layer GNN, **it is unclear how effectively the model addresses the oversmoothing issue**. A more convincing approach would be to experimentally demonstrate that oversmoothing is a problem for other baseline models (e.g., by comparing node similarity across embeddings from different layers) rather than making direct assertions. Furthermore, a similar similarity analysis should be conducted for DHENN to substantiate this claim.
-  Regarding limitation 3 (Non end-to-end framework): Many existing methods, whether based on single-graph or knowledge graph approaches, are already end-to-end. Thus, it is **difficult to convincingly argue that this is a major limitation** in the current landscape of DDI prediction research.

**Questions:**

- Could you please define what constitutes a "hybrid method" in the context of your study and elaborate on how it differs from KG-based methods? Typically, KG-based methods also employ end-to-end node representation learning or feature extraction.
- What specific modalities are used in DHENN to establish a multi-modal framework? It appears that molecular structures are not used directly but are instead converted into molecular fingerprints.
- How are substructures extracted and selected from drug SMILES?
- MACCS are typically 166-dimensional. Why does DHENN only utilize 13 MACCS features?
- The necessity for neighborhood sampling in DHENN is unclear. In general, hetero-GNNs, similar to the one used in your model, often consider all neighbors to preserve the complete network topology.
- Could you clarify the difference between Task 2 and Task 3?
- The experimental settings for the non-end-to-end version of DHENN lack. Specifically, what was the optimization objective for the feature extraction component in this configuration?

---

> ### Author Response · Authors · 2024-11-25
>
> We thank the reviewer for the insightful and constructive feedback. Below, we address each weakness and question raised one by one.
>
> +++ Regarding **Weaknesses** +++
>
> **W1**: High-order interactions.
>
> **Q1**: We acknowledge the concern about the single-layer GNN capturing only first-order relationships. However, the constructed Multimodal Knowledge Graph (MKG) in our framework inherently encodes high-order interactions by integrating four types of drug-centered binary relationships: <drugs, chemical entities>, <drugs, substructures> (coded by SMILES), <drugs, drugs> (representing drug-drug interactions), and <drugs, molecular fingerprints> (coded by MACCS). This topology enables DHENN to capture complex, high-order relationships without requiring multiple GNN layers. Note, while stacking GNN layers could propagate higher-order interactions, it often results in diminished model performance due to oversmoothing, which we address directly in our next point.
>
> **W2**: Oversmoothing.
>
> **Q2**: We have demonstrated this in Figure 3, which shows that increasing GNN layers decreases performance. In contrast, our PECN model component enhances performance as its depth increases, validating its ability to mitigate oversmoothing. Prior works like MKG-FENN (Wu et al., 2024) and MDNN (Lyu et al., 2021) also rely on single-layer GNNs for the same reason, as discussed in their original papers.
>
> **W3**: Non end-to-end framework design.
>
> **A3**: We acknowledge that some existing methods are end-to-end. However, note that these methods often rely on incomplete or less comprehensive knowledge graphs, limiting their predictive power. In contrast, DHENN constructs a multimodal and comprehensive MKG, which enhances its predictive capacity. The end-to-end integration of this enriched graph structure ensures that representation learning and prediction are jointly optimized, yielding better performance. We will clarify this and refine our discussion of this limitation in the camera-ready.
>
> +++ Regarding **Questions** +++
>
> **Q1**:  What constitutes a "hybrid method"  and how does it differ from KG-based methods?
>
> **A1**:  DHENN is termed "hybrid" because it combines two distinct neural networks: a GNN for learning KG representations, and the Prediction-Enhanced Cascading Network (PECN) for DDI prediction. In contrast, other KG-based methods often separate drug-centered binary relations, neglecting high-order DDI interactions. In addition, the existing KG-based methods tend to use shallow architectures to avoid oversmoothing, whereas our PECN network component allows for stacking deeper GCN layers to enhance learning efficacy.
>
> **Q2**:  What specific modalities are used in DHENN to establish a multimodal framework?
>
> **A2**: Our MKG encodes high-order interactions by integrating four types of drug-centered binary relationships: <drugs, chemical entities>, <drugs, substructures> (coded by SMILES), <drugs, drugs> (representing drug-drug interactions), and <drugs, molecular fingerprints> (coded by MACCS).
>
> **Q3**:  How are substructures extracted and selected from drug SMILES?
>
> **A3**: We followed prior studies such as Veber et al. (2002) and Baranwal et al. (2020) to select 13 MACCS key features and 7 additional molecular features, ensuring the relevance and robustness of the selected features.
>
> **Q4**:  Why is neighborhood sampling necessary in DHENN?
>
> **A4**: As shown in Figure 6 (Appendix), neighborhood sampling balances efficiency and noise handling. Small neighborhood sizes hinder information propagation, while large sizes amplify noise sensitivity. Our approach ensures optimal performance by balancing these trade-offs.
>
> **Q5**:  What is the difference between Task 2 and Task 3?
>
> **A5**:  We elaborate Tasks 1, 2, and 3 for better comparisons. Task 1 assumes all drugs are known during training, Task 2 introduces settings where only partial drugs are known, and Task 3 evaluates the model’s ability to generalize to entirely new drugs unseen during training. These tasks reflect progressively more challenging scenarios, highlighting DHENN’s robustness in generalization.
>
> **Q6**:  What are the experimental settings and optimization objectives for the non-end-to-end version of DHENN?
>
> **A6**:  We  implemented  the sequential two-phase training  with distinct optimization objectives, where the GNN phase optimized the reconstruction loss of graph structures, and the PECN phase optimized the prediction loss for DDI events. In contrast, the end-to-end version of DHENN unifies these objectives under a single framework (Equation 9), ensuring cohesive optimization across all components and ultimately improving performance. This unified optimization process is key to DHENN’s superior performance compared to the non-end-to-end version.

---

> > ### Comment · Reviewer_tLVQ · 2024-11-26
> > **Responses to the authors**
> >
> > Thank you for the well-prepared responses that address some of my concerns. However, I would maintain some of my concerns since previous responses are not that convincing:
> >
> > +++ **Regarding Weaknesses** +++
> >
> > **1. High-Order Interactions**: The concept of "high-order information" as presented in this study may be misunderstood. The drug-centered relationships described seem more like primary-order information related to biological functions or chemical properties, rather than high-order interactions. A better example of high-order information would be drug-drug interactions inferred through shared neighboring nodes, which typically require multi-layered GNNs to capture.
> >
> > **2. Oversmoothing**: As previously mentioned, the DHENN model does not effectively address the issue of oversmoothing, which is a phenomenon caused by deeper GNNs. The results in Figure 3 further support this, as the performance of DHENN is significantly degraded when deeper GNNs are applied. The proposed predictor/decoder may improve prediction accuracy compared to MLPs, but it does not directly resolve the oversmoothing problem as defined in your cited papers (Chen et al., 2020; Liu et al., 2020).
> >
> > **3. Non end-to-end problem**: As acknowledged in your response, there are existing end-to-end methods, which makes this limitation less convincing as a major drawback of previous studies. A more impactful challenge might be the incomplete nature of the KGs in previous studies, as described in your response. This could be framed as a more pressing limitation to address.
> > In summary, there appear to be some inconsistencies between the introduced limitations and challenges of current DDI prediction and how they are addressed in the proposed method. Further investigation and clarification are recommended to strengthen the manuscript.
> >
> > +++ **Regarding Questions** +++
> >
> > **1. Modalities used**: Can binary relational data truly be regarded as distinct modalities for representation learning?
> >
> > **2. MACCS features**: The original study by Baranwal et al. (2020) focused on metabolic pathway prediction, which seems only tangentially related to drug-drug interaction prediction. The rationale for selecting a subset of MACCS features based on that study is debatable, given the potential lack of transferable knowledge between the two tasks.

---

> ### Author Response · Authors · 2024-11-28
>
> We thank the reviewer for the discussions and for providing us the opportunity to further clarify your concerns. Following your comments, we have revised the inappropriate claims and descriptions. We address the raised concerns point by point as follows.
>
>
>
> +++ Regarding **Weaknesses** +++
>
> **W1**: High-Order Interactions.
>
> **A1**: Thank you for pointing this issue out. We agree that the drug-centered multiple binary relationships associated with chemical entities, substructures, and molecular fingerprints mostly represent the high-order “information” instead of “interactions” for DDI prediction. We coin it as “high-order” because, comparing with other DDI methods, our MKG design includes multiple different types of drug-centered relations. We agree that coining such diverse topological information captured by MKG as “high-order interaction” may impose comprehension challenges and confusions. We have revised this term as “high-order information” throughout the manuscript. Please refer to the updated manuscript.
>
>
>
>
> **W2**: Oversmoothing.
>
> **A2**: Oversmoothing of GNNs (Chen et al. 2020) in DDI events prediction is associated with two aspects, i.e., the message-passing (via edges) and the structural feature extraction (from nodes) on a drug-centered knowledge graph. We would like to kindly note that, in previous studies (e.g., Wu et al., 2024, Liu et al., 2021), their GNNs remain shallow not only in the message-passing layers but also in the representation (fully-connected) layers due to oversmoothing. Indeed, our DHENN does not address the oversmoothing issue associated with message-passing in DDI events prediction. However, by designing the deeper PECN, our design mitigates the oversmoothing associated with the nodal feature extraction in DDI events prediction. As shown in Figure 3, increasing the depth of PECN can boost performance. We agree that claiming that DHENN can address oversmoothing in GNNs in general is not fully accurate. We have clarified that our paper addresses the oversmoothing issue associated with feature extraction by designing PECN that allows for deeper layers in the updated manuscript. We acknowledge that the oversmoothing issue associated with message-passing remains an open problem that we will discuss in the camera ready.
>
> [1] Chen et al., Measuring and relieving the oversmoothing problem for graph neural networks from the topological view. AAAI 2020.
>
> [2] Wu et al., Mkg-fenn: A multimodal knowledge graph fused end-to-end neural network for accurate drug–drug interaction prediction. AAAI 2024.
>
> [3] Tengfei Lyu, Jianliang Gao, Ling Tian, Zhao Li, Peng Zhang, and Ji Zhang. Mdnn: A multimodal deep neural network for predicting drug-drug interaction events. In IJCAI 2021
>
>
>
>
>
> **W3**: Non end-to-end problem.
>
> **A3**: We appreciate this suggestion. You are correct that non-end-to-end is not the major drawback of previous studies. The main challenge of current DDI prediction is how to comprehensively explore drug-centered KGs to learn a unified predictor. Previous studies either partially consider drug-centered KGs or do not unify multiple KGs into an end-to-end learning model. To this end, our DHENN constructs the unified MKG that includes drug-centered multiple KGs. Such MKG is learned by an end-to-end learning way. Table 4 (manuscript) and Table 5 (Appendix) have validated the effectiveness of multiple KGs. Figure 4 has verified the performance improvement by an end-to-end learning way. Following your suggestion, we have clarified and revised the corresponding introduced limitations and challenges of current DDI prediction in the updated manuscript.
>
>
>
> +++ Regarding **Questions** +++
>
> **Q1**: Modalities used.
>
> **A1**: The considered four types of drug-centered binary relationships are: <drugs, chemical entities>, <drugs, substructures> (coding by SMILES), <drugs, drugs> (i.e., DDI events), and <drugs, molecular fingerprints>(coding by MACCS). They are different binary relationships. Table 4 (manuscript) and Table 5 (Appendix) have validated the effectiveness of each type of binary relationship.
>
>
>
> **Q2**: MACCS features.
>
> **A2**: Thanks for pointing this issue out. One important mechanism of DDIs prediction is to discover the similar characteristics of drugs. Although the metabolic pathway and drug-drug interaction predictions lack transferable knowledge, the MACCS features can still represent some characteristics of drugs, which is benificial for finding the similar characteristics of drugs. Table 4 (manuscript) and Table 5 (Appendix) have validated the effectiveness of MACCS features (i.e., P4 ). In the furture, we will investigate the selecting issue of MACCS features following your suggestion.

---

### Official Review · Reviewer_nZSN · 2024-11-04

**Soundness:** 3
**Presentation:** 2
**Contribution:** 3
**Rating:** 6
**Confidence:** 5

**Summary:**

In this paper, the authors proposed a new framework DHENN which integrates the different entities related to molecule, and perform drug-drug interaction events prediction. In order to utilize these entites information, the authors propsed multimodal knowledge graph (MKG) and prediction enhanced cascading network (PECN). In two datasets and different tasks, DHENN outperforms other methods.

**Strengths:**

1. The experiments are thorough. The authors performed the experiments on both DDIMDL's dataset and a recently-collected dataset.
2. I doubted the architecture of the prediction enhanced cascading network (PECN) at first, but the authors used ablation study to prove the effectiveness of the PECN, which is impressing.

**Weaknesses:**

The quality of presentation can still be improved, there are many problems such as:
1. For Figure 1, target, transporter and enzyme. They are all proteins. But why did the authors use atomic nucleus icon to represent target? And for the enzyme, I believed the icon is bacteria.
2. Also in Figure 1, please use molecule SMILES to represent SMILES. For example, C1=CC=CC=C1 for Benzene. And I don't know what does the authors want to do by putting two molecule graphs into drug SMILES and repeate it again. Is it representing $drug_i$ and $drug_j$?
3. In the drug-substructures, it said the drug has a SMILES attribute “986”, what does it mean? That's so confusing. And also, the authors didn't introduce SMILES at all.

**Questions:**

See above

---

> ### Author Response · Authors · 2024-11-25
>
> We thank the reviewer for the encouraging and constructive comments, where we address your concerns point by point below.
>
> **Q1**: Icons used in Figure 1 were inappropriate.
>
> **A1**: Thank you for pointing out these issues with Figure 1. We have uploaded the updated Figure 1, which now uses correct and standardized icons to represent targets, transporters, and enzymes as proteins.
>
>
>
> **Q2**: SMILES representation, and why two molecule graphs in Figure 1.
>
> **A2**: We have revised Figure 1 to represent SMILES using actual SMILES strings to provide a more accurate depiction, as you suggested. Regarding the two molecule graphs, they were intended to represent two different drugs, drug$_i$ and drug$_j$, involved in a drug-drug interaction. We acknowledge that this was not clearly conveyed in the original figure. In the revised version, we have explicitly labeled the molecules as drug$_i$ and drug$_j$ to clarify their roles within our framework. This should help readers better understand the context and purpose of the illustration.
>
> **Q3**: What does the SMILES attribute “986” refer to? Why not mentioning SMILES?
>
> **A3**: The number “986” represents a molecular fingerprint, which is a numerical descriptor used to encode chemical properties. For example, the triplet <Lovastatin, includes, 986> indicates that the drug Lovastatin contains the substructure corresponding to bit “986” in its molecular fingerprint, where the value "986" refers to a unique molecular fingerprint generated using the Morgan Fingerprint method (also known as ECFP) from the RDKit library [1]. To further clarify, this process include 1) generating the drug molecular information, including the SMILES string, drug ID, target, and enzyme data., 2) converting the SMILES string into a molecular object using the "Chem.MolFromSmiles" method from RDKit, and 3) using the "AllChem.GetMorganFingerprintAsBitVect" method to generate the molecular fingerprint from valid molecular objects.
>
> To aid readers who do not have a background in molecular research, we have introduced the concept of SMILES in the revised manuscript to avoid ambiguity. Specifically, SMILES (Simplified Molecular Input Line Entry System) is a text-based notation system that represents chemical structures. It uses atomic symbols (e.g., C, O, H) and bond types (e.g., - for single bonds, = for double bonds) alongside parentheses for branching and numbers for ring structures. SMILES provides a compact way to describe molecules, making it easy to store and share chemical information. We will supplement this description in our camera ready.
>
>
> [1] https://zenodo.org/records/13990314

---

### Meta-Review · Area_Chair_1BiJ · 2024-12-17

**Metareview:**

This paper improves DDI prediction by capturing high-order interactions, and reducing over-smoothing. However, as Reviewer FpBb mentioned, the experimental results suggest that increasing the depth of PECN does not result in significant improvements. Additionally, oversmoothing does not appear to be a major concern for DDI prediction, as comparable performance can be achieved with shallow networks.

**Additional Comments On Reviewer Discussion:**

While the authors have provided responses, they have not adequately addressed the related concerns (Reviewer FpBb).

---

### Decision · Program_Chairs · 2025-01-22

Reject